# Deep learning with Elastic Averaging SGD

**Sixin Zhang**
Courant Institute, NYU
zsx@cims.nyu.edu

**Anna Choromanska**
Courant Institute, NYU
achoroma@cims.nyu.edu

**Yann LeCun**
Center for Data Science, NYU & Facebook AI Research
yann@cims.nyu.edu

## Abstract

We study the problem of stochastic optimization for deep learning in the parallel computing environment under communication constraints. A new algorithm is proposed in this setting where the communication and coordination of work among concurrent processes (local workers), is based on an elastic force which links the parameters they compute with a center variable stored by the parameter server (master). The algorithm enables the local workers to perform more exploration, i.e. the algorithm allows the local variables to fluctuate further from the center variable by reducing the amount of communication between local workers and the master. We empirically demonstrate that in the deep learning setting, due to the existence of many local optima, allowing more exploration can lead to the improved performance. We propose synchronous and asynchronous variants of the new algorithm. We provide the stability analysis of the asynchronous variant in the round-robin scheme and compare it with the more common parallelized method *ADMM*. We show that the stability of *EASGD* is guaranteed when a simple stability condition is satisfied, which is not the case for *ADMM*. We additionally propose the momentum-based version of our algorithm that can be applied in both synchronous and asynchronous settings. Asynchronous variant of the algorithm is applied to train convolutional neural networks for image classification on the *CIFAR* and *ImageNet* datasets. Experiments demonstrate that the new algorithm accelerates the training of deep architectures compared to *DOWNPOUR* and other common baseline approaches and furthermore is very communication efficient.

## 1 Introduction

One of the most challenging problems in large-scale machine learning is how to parallelize the training of large models that use a form of stochastic gradient descent (*SGD*) [1]. There have been attempts to parallelize *SGD*-based training for large-scale deep learning models on large number of CPUs, including the Google's Distbelief system [2]. But practical image recognition systems consist of large-scale convolutional neural networks trained on few GPU cards sitting in a single computer [3, 4]. The main challenge is to devise parallel *SGD* algorithms to train large-scale deep learning models that yield a significant speedup when run on multiple GPU cards.

In this paper we introduce the *Elastic Averaging SGD* method (*EASGD*) and its variants. *EASGD* is motivated by quadratic penalty method [5], but is re-interpreted as a parallelized extension of the averaging *SGD* algorithm [6]. The basic idea is to let each worker maintain its own local parameter, and the communication and coordination of work among the local workers is based on an elastic force which links the parameters they compute with a center variable stored by the master. The center variable is updated as a moving average where the average is taken in time and also in space over the parameters computed by local workers. The main contribution of this paper is a new algorithm that provides fast convergent minimization while outperforming *DOWNPOUR* method [2] and other

baseline approaches in practice. Simultaneously it reduces the communication overhead between the master and the local workers while at the same time it maintains high-quality performance measured by the test error. The new algorithm applies to deep learning settings such as parallelized training of convolutional neural networks.

The article is organized as follows. Section 2 explains the problem setting, Section 3 presents the synchronous *EASGD* algorithm and its asynchronous and momentum-based variants, Section 4 provides stability analysis of *EASGD* and *ADMM* in the round-robin scheme, Section 5 shows experimental results and Section 6 concludes. The Supplement contains additional material including additional theoretical analysis.

## 2   Problem setting

Consider minimizing a function $F(x)$ in a parallel computing environment [7] with $p \in \mathbb{N}$ workers and a master. In this paper we focus on the stochastic optimization problem of the following form

$$\min_x F(x) := \mathbb{E}[f(x, \xi)], \tag{1}$$

where $x$ is the model parameter to be estimated and $\xi$ is a random variable that follows the probability distribution $\mathbb{P}$ over $\Omega$ such that $F(x) = \int_\Omega f(x, \xi) \mathbb{P}(d\xi)$. The optimization problem in Equation 1 can be reformulated as follows

$$\min_{x^1, \dots, x^p, \tilde{x}} \sum_{i=1}^p \mathbb{E}[f(x^i, \xi^i)] + \frac{\rho}{2} \|x^i - \tilde{x}\|^2, \tag{2}$$

where each $\xi^i$ follows the same distribution $\mathbb{P}$ (thus we assume each worker can sample the entire dataset). In the paper we refer to $x^i$'s as local variables and we refer to $\tilde{x}$ as a center variable. The problem of the equivalence of these two objectives is studied in the literature and is known as the *augmentability* or the *global variable consensus* problem [8, 9]. The quadratic penalty term $\rho$ in Equation 2 is expected to ensure that local workers will not fall into different attractors that are far away from the center variable. This paper focuses on the problem of reducing the parameter communication overhead between the master and local workers [10, 2, 11, 12, 13]. The problem of data communication when the data is distributed among the workers [7, 14] is a more general problem and is not addressed in this work. We however emphasize that our problem setting is still highly non-trivial under the communication constraints due to the existence of many local optima [15].

## 3   EASGD update rule

The *EASGD* updates captured in resp. Equation 3 and 4 are obtained by taking the gradient descent step on the objective in Equation 2 with respect to resp. variable $x^i$ and $\tilde{x}$,

$$x_{t+1}^i = x_t^i - \eta(g_t^i(x_t^i) + \rho(x_t^i - \tilde{x}_t)) \tag{3}$$

$$\tilde{x}_{t+1} = \tilde{x}_t + \eta \sum_{i=1}^p \rho(x_t^i - \tilde{x}_t), \tag{4}$$

where $g_t^i(x_t^i)$ denotes the stochastic gradient of $F$ with respect to $x^i$ evaluated at iteration $t$, $x_t^i$ and $\tilde{x}_t$ denote respectively the value of variables $x^i$ and $\tilde{x}$ at iteration $t$, and $\eta$ is the learning rate.

The update rule for the center variable $\tilde{x}$ takes the form of moving average where the average is taken over both space and time. Denote $\alpha = \eta\rho$ and $\beta = p\alpha$, then Equation 3 and 4 become

$$x_{t+1}^i = x_t^i - \eta g_t^i(x_t^i) - \alpha(x_t^i - \tilde{x}_t) \tag{5}$$

$$\tilde{x}_{t+1} = (1 - \beta)\tilde{x}_t + \beta \left( \frac{1}{p} \sum_{i=1}^p x_t^i \right). \tag{6}$$

Note that choosing $\beta = p\alpha$ leads to an elastic symmetry in the update rule, i.e. there exists an symmetric force equal to $\alpha(x_t^i - \tilde{x}_t)$ between the update of each $x^i$ and $\tilde{x}$. It has a crucial influence on the algorithm's stability as will be explained in Section 4. Also in order to minimize the staleness [16] of the difference $x_t^i - \tilde{x}_t$ between the center and the local variable, the update for the master in Equation 4 involves $x_t^i$ instead of $x_{t+1}^i$.

Note also that $\alpha = \eta\rho$, where the magnitude of $\rho$ represents the amount of exploration we allow in the model. In particular, small $\rho$ allows for more exploration as it allows $x^i$'s to fluctuate further from the center $\tilde{x}$. The distinctive idea of *EASGD* is to allow the local workers to perform more exploration (small $\rho$) and the master to perform exploitation. This approach differs from other settings explored in the literature [2, 17, 18, 19, 20, 21, 22, 23], and focus on how fast the center variable converges. In this paper we show the merits of our approach in the deep learning setting.

## 3.1 Asynchronous EASGD

We discussed the synchronous update of *EASGD* algorithm in the previous section. In this section we propose its asynchronous variant. The local workers are still responsible for updating the local variables $x^i$'s, whereas the master is updating the center variable $\tilde{x}$. Each worker maintains its own clock $t^i$, which starts from 0 and is incremented by 1 after each stochastic gradient update of $x^i$ as shown in Algorithm 1. The master performs an update whenever the local workers finished $\tau$ steps of their gradient updates, where we refer to $\tau$ as the *communication period*. As can be seen in Algorithm 1, whenever $\tau$ divides the local clock of the $i^{\text{th}}$ worker, the $i^{\text{th}}$ worker communicates with the master and requests the current value of the center variable $\tilde{x}$. The worker then waits until the master sends back the requested parameter value, and computes the elastic difference $\alpha(x - \tilde{x})$ (this entire procedure is captured in step a) in Algorithm 1). The elastic difference is then sent back to the master (step b) in Algorithm 1) who then updates $\tilde{x}$.

The communication period $\tau$ controls the frequency of the communication between every local worker and the master, and thus the trade-off between exploration and exploitation.

| **Algorithm 1:** Asynchronous EASGD: Processing by worker $i$ and the master | **Algorithm 2:** Asynchronous EAMSGD: Processing by worker $i$ and the master |
|---|---|
| **Input:** learning rate $\eta$, moving rate $\alpha$, communication period $\tau \in \mathbb{N}$ <br> **Initialize:** $\tilde{x}$ is initialized randomly, $x^i = \tilde{x}$, $t^i = 0$ | **Input:** learning rate $\eta$, moving rate $\alpha$, communication period $\tau \in \mathbb{N}$, momentum term $\delta$ <br> **Initialize:** $\tilde{x}$ is initialized randomly, $x^i = \tilde{x}$, $v^i = 0$, $t^i = 0$ |
| **Repeat** <br>    $x \leftarrow x^i$ <br>    **if** ($\tau$ divides $t^i$) **then** <br>       **a)** $x^i \leftarrow x^i - \alpha(x - \tilde{x})$ <br>       **b)** $\tilde{x} \leftarrow \tilde{x} + \alpha(x - \tilde{x})$ <br>    **end** <br>    $x^i \leftarrow x^i - \eta g^i_{t^i}(x)$ <br>    $t^i \leftarrow t^i + 1$ <br> **Until forever** | **Repeat** <br>    $x \leftarrow x^i$ <br>    **if** ($\tau$ divides $t^i$) **then** <br>       **a)** $x^i \leftarrow x^i - \alpha(x - \tilde{x})$ <br>       **b)** $\tilde{x} \leftarrow \tilde{x} + \alpha(x - \tilde{x})$ <br>    **end** <br>    $v^i \leftarrow \delta v^i - \eta g^i_{t^i}(x + \delta v^i)$ <br>    $x^i \leftarrow x^i + v^i$ <br>    $t^i \leftarrow t^i + 1$ <br> **Until forever** |

## 3.2 Momentum EASGD

The momentum EASGD (*EAMSGD*) is a variant of our Algorithm 1 and is captured in Algorithm 2. It is based on the Nesterov's momentum scheme [24, 25, 26], where the update of the local worker of the form captured in Equation 3 is replaced by the following update

$$
\begin{aligned}
v^i_{t+1} &= \delta v^i_t - \eta g^i_t(x^i_t + \delta v^i_t) \\
x^i_{t+1} &= x^i_t + v^i_{t+1} - \eta\rho(x^i_t - \tilde{x}_t),
\end{aligned} \tag{7}
$$

where $\delta$ is the momentum term. Note that when $\delta = 0$ we recover the original *EASGD* algorithm.

As we are interested in reducing the communication overhead in the parallel computing environment where the parameter vector is very large, we will be exploring in the experimental section the asynchronous *EASGD* algorithm and its momentum-based variant in the relatively large $\tau$ regime (less frequent communication).

# 4 Stability analysis of EASGD and ADMM in the round-robin scheme

In this section we study the stability of the asynchronous *EASGD* and *ADMM* methods in the round-robin scheme [20]. We first state the updates of both algorithms in this setting, and then we study

their stability. We will show that in the one-dimensional quadratic case, *ADMM* algorithm can exhibit chaotic behavior, leading to exponential divergence. The analytic condition for the *ADMM* algorithm to be stable is still unknown, while for the *EASGD* algorithm it is very simple[1].

The analysis of the synchronous *EASGD* algorithm, including its convergence rate, and its averaging property, in the quadratic and strongly convex case, is deferred to the Supplement.

In our setting, the *ADMM* method [9, 27, 28] involves solving the following minimax problem[2],

$$\max_{\lambda^1,\ldots,\lambda^p} \min_{x^1,\ldots,x^p,\tilde{x}} \sum_{i=1}^p F(x^i) - \lambda^i(x^i - \tilde{x}) + \frac{\rho}{2}\|x^i - \tilde{x}\|^2, \qquad (8)$$

where $\lambda^i$'s are the Lagrangian multipliers. The resulting updates of the *ADMM* algorithm in the round-robin scheme are given next. Let $t \geq 0$ be a global clock. At each $t$, we linearize the function $F(x^i)$ with $F(x_t^i) + \langle \nabla F(x_t^i), x^i - x_t^i \rangle + \frac{1}{2\eta}\|x^i - x_t^i\|^2$ as in [28]. The updates become

$$\lambda_{t+1}^i = \begin{cases} \lambda_t^i - (x_t^i - \tilde{x}_t) & \text{if} \quad \mod(t,p) = i - 1; \\ \lambda_t^i & \text{if} \quad \mod(t,p) \neq i - 1. \end{cases} \qquad (9)$$

$$x_{t+1}^i = \begin{cases} \frac{x_t^i - \eta\nabla F(x_t^i) + \eta\rho(\lambda_{t+1}^i + \tilde{x}_t)}{1+\eta\rho} & \text{if} \quad \mod(t,p) = i - 1; \\ x_t^i & \text{if} \quad \mod(t,p) \neq i - 1. \end{cases} \qquad (10)$$

$$\tilde{x}_{t+1} = \frac{1}{p}\sum_{i=1}^p (x_{t+1}^i - \lambda_{t+1}^i). \qquad (11)$$

Each local variable $x^i$ is periodically updated (with period $p$). First, the Lagrangian multiplier $\lambda^i$ is updated with the dual ascent update as in Equation 9. It is followed by the gradient descent update of the local variable as given in Equation 10. Then the center variable $\tilde{x}$ is updated with the most recent values of all the local variables and Lagrangian multipliers as in Equation 11. Note that since the step size for the dual ascent update is chosen to be $\rho$ by convention [9, 27, 28], we have re-parametrized the Lagrangian multiplier to be $\lambda_t^i \leftarrow \lambda_t^i/\rho$ in the above updates.

The *EASGD* algorithm in the round-robin scheme is defined similarly and is given below

$$x_{t+1}^i = \begin{cases} x_t^i - \eta\nabla F(x_t^i) - \alpha(x_t^i - \tilde{x}_t) & \text{if} \quad \mod(t,p) = i - 1; \\ x_t^i & \text{if} \quad \mod(t,p) \neq i - 1. \end{cases} \qquad (12)$$

$$\tilde{x}_{t+1} = \tilde{x}_t + \sum_{i:\ \mod(t,p)=i-1} \alpha(x_t^i - \tilde{x}_t). \qquad (13)$$

At time $t$, only the $i$-th local worker (whose index $i-1$ equals $t$ modulo $p$) is activated, and performs the update in Equations 12 which is followed by the master update given in Equation 13.

We will now focus on the one-dimensional quadratic case without noise, i.e. $F(x) = \frac{x^2}{2}, x \in \mathbb{R}$.

For the *ADMM* algorithm, let the state of the (dynamical) system at time $t$ be $s_t = (\lambda_t^1, x_t^1, \ldots, \lambda_t^p, x_t^p, \tilde{x}_t) \in \mathbb{R}^{2p+1}$. The local worker $i$'s updates in Equations 9, 10, and 11 are composed of three linear maps which can be written as $s_{t+1} = (F_3^i \circ F_2^i \circ F_1^i)(s_t)$. For simplicity, we will only write them out below for the case when $i = 1$ and $p = 2$:

$$F_1^1 = \begin{pmatrix} 1 & -1 & 0 & 0 & 1 \\ 0 & 1 & 0 & 0 & 0 \\ 0 & 0 & 1 & 0 & 0 \\ 0 & 0 & 0 & 1 & 0 \\ 0 & 0 & 0 & 0 & 1 \end{pmatrix}, F_2^1 = \begin{pmatrix} 1 & 0 & 0 & 0 & 0 \\ \frac{\eta\rho}{1+\eta\rho} & \frac{1-\eta}{1+\eta\rho} & 0 & 0 & \frac{\eta\rho}{1+\eta\rho} \\ 0 & 0 & 1 & 0 & 0 \\ 0 & 0 & 0 & 1 & 0 \\ 0 & 0 & 0 & 0 & 1 \end{pmatrix}, F_3^1 = \begin{pmatrix} 1 & 0 & 0 & 0 & 0 \\ 0 & 1 & 0 & 0 & 0 \\ 0 & 0 & 1 & 0 & 0 \\ 0 & 0 & 0 & 1 & 0 \\ -\frac{1}{p} & \frac{1}{p} & -\frac{1}{p} & \frac{1}{p} & 0 \end{pmatrix}.$$

For each of the $p$ linear maps, it's possible to find a simple condition such that each map, where the $i^{\text{th}}$ map has the form $F_3^i \circ F_2^i \circ F_1^i$, is stable (the absolute value of the eigenvalues of the map are

smaller or equal to one). However, when these non-symmetric maps are composed one after another as follows $\mathcal{F} = F_3^p \circ F_2^p \circ F_1^p \circ \ldots \circ F_3^1 \circ F_2^1 \circ F_1^1$, the resulting map $\mathcal{F}$ can become unstable! (more precisely, some eigenvalues of the map can sit outside the unit circle in the complex plane).

We now present the numerical conditions for which the *ADMM* algorithm becomes unstable in the round-robin scheme for $p = 3$ and $p = 8$, by computing the largest absolute eigenvalue of the map $\mathcal{F}$. Figure 1 summarizes the obtained result.

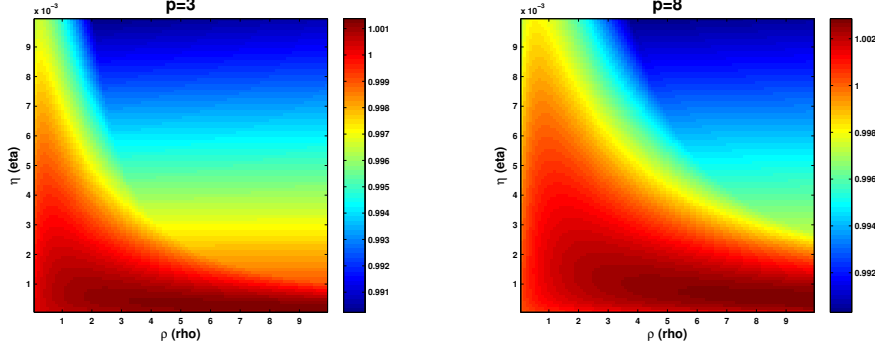

Figure 1: The largest absolute eigenvalue of the linear map $\mathcal{F} = F_3^p \circ F_2^p \circ F_1^p \circ \ldots \circ F_3^1 \circ F_2^1 \circ F_1^1$ as a function of $\eta \in (0, 10^{-2})$ and $\rho \in (0, 10)$ when $p = 3$ and $p = 8$. To simulate the chaotic behavior of the *ADMM* algorithm, one may pick $\eta = 0.001$ and $\rho = 2.5$ and initialize the state $s_0$ either randomly or with $\lambda_0^i = 0, x_0^i = \tilde{x}_0 = 1000, \forall i$. *Figure should be read in color.*

On the other hand, the *EASGD* algorithm involves composing only symmetric linear maps due to the elasticity. Let the state of the (dynamical) system at time $t$ be $s_t = (x_t^1, \ldots, x_t^p, \tilde{x}_t) \in \mathbb{R}^{p+1}$. The activated local worker $i$'s update in Equation 12 and the master update in Equation 13 can be written as $s_{t+1} = F^i(s_t)$. In case of $p = 2$, the map $F^1$ and $F^2$ are defined as follows

$$F^1 = \begin{pmatrix} 1 - \eta - \alpha & 0 & \alpha \\ 0 & 1 & 0 \\ \alpha & 0 & 1 - \alpha \end{pmatrix}, F^2 = \begin{pmatrix} 1 & 0 & 0 \\ 0 & 1 - \eta - \alpha & \alpha \\ 0 & \alpha & 1 - \alpha \end{pmatrix}$$

For the composite map $F^p \circ \ldots \circ F^1$ to be stable, the condition that needs to be satisfied is actually the same for each $i$, and is furthermore independent of $p$ (since each linear map $F^i$ is symmetric). It essentially involves the stability of the $2 \times 2$ matrix $\begin{pmatrix} 1 - \eta - \alpha & \alpha \\ \alpha & 1 - \alpha \end{pmatrix}$, whose two (real) eigenvalues $\lambda$ satisfy $(1 - \eta - \alpha - \lambda)(1 - \alpha - \lambda) = \alpha^2$. The resulting stability condition ($|\lambda| \leq 1$) is simple and given as $0 \leq \eta \leq 2, 0 \leq \alpha \leq \frac{4 - 2\eta}{4 - \eta}$.

## 5  Experiments

In this section we compare the performance of *EASGD* and *EAMSGD* with the parallel method *DOWNPOUR* and the sequential method *SGD*, as well as their averaging and momentum variants.

All the parallel comparator methods are listed below[3]:

- *DOWNPOUR* [2], the pseudo-code of the implementation of *DOWNPOUR* used in this paper is enclosed in the Supplement.
- *Momentum DOWNPOUR* (*MDOWNPOUR*), where the Nesterov's momentum scheme is applied to the master's update (note it is unclear how to apply it to the local workers or for the case when $\tau > 1$). The pseudo-code is in the Supplement.
- A method that we call *ADOWNPOUR*, where we compute the average over time of the center variable $\tilde{x}$ as follows: $z_{t+1} = (1 - \alpha_{t+1})z_t + \alpha_{t+1}\tilde{x}_t$, and $\alpha_{t+1} = \frac{1}{t+1}$ is a moving rate, and $z_0 = \tilde{x}_0$. $t$ denotes the master clock, which is initialized to 0 and incremented every time the center variable $\tilde{x}$ is updated.
- A method that we call *MVADOWNPOUR*, where we compute the moving average of the center variable $\tilde{x}$ as follows: $z_{t+1} = (1 - \alpha)z_t + \alpha\tilde{x}_t$, and the moving rate $\alpha$ was chosen to be constant, and $z_0 = \tilde{x}_0$. $t$ denotes the master clock and is defined in the same way as for the *ADOWNPOUR* method.

All the sequential comparator methods ($p = 1$) are listed below:

- *SGD* [1] with constant learning rate $\eta$.
- *Momentum SGD* (*MSGD*) [26] with constant momentum $\delta$.
- *ASGD* [6] with moving rate $\alpha_{t+1} = \frac{1}{t+1}$.
- *MVASGD* [6] with moving rate $\alpha$ set to a constant.

We perform experiments in a deep learning setting on two benchmark datasets: CIFAR-10 (we refer to it as *CIFAR*) [4] and ImageNet ILSVRC 2013 (we refer to it as *ImageNet*) [5]. We focus on the image classification task with deep convolutional neural networks. We next explain the experimental setup. The details of the data preprocessing and prefetching are deferred to the Supplement.

## 5.1 Experimental setup

For all our experiments we use a GPU-cluster interconnected with InfiniBand. Each node has 4 Titan GPU processors where each local worker corresponds to one GPU processor. The center variable of the master is stored and updated on the centralized parameter server [2][6].

To describe the architecture of the convolutional neural network, we will first introduce a notation. Let $(c, y)$ denotes the size of the input image to each layer, where $c$ is the number of color channels and $y$ is both the horizontal and the vertical dimension of the input. Let $C$ denotes the fully-connected convolutional operator and let $P$ denotes the max pooling operator, $D$ denotes the linear operator with dropout rate equal to $0.5$ and $S$ denotes the linear operator with softmax output non-linearity. We use the cross-entropy loss and all inner layers use rectified linear units. For the *ImageNet* experiment we use the similar approach to [4] with the following 11-layer convolutional neural network (3,221)C(96,108)P(96,36)C(256,32)P(256,16)C(384,14) C(384,13)C(256,12)P(256,6)D(4096,1)D(4096,1)S(1000,1). For the *CIFAR* experiment we use the similar approach to [29] with the following 7-layer convolutional neural network (3,28)C(64,24)P(64,12)C(128,8)P(128,4)C(64,2)D(256,1)S(10,1).

In our experiments all the methods we run use the same initial parameter chosen randomly, except that we set all the biases to zero for *CIFAR* case and to 0.1 for *ImageNet* case. This parameter is used to initialize the master and all the local workers[7]. We add $l_2$-regularization $\frac{\lambda}{2} \|x\|^2$ to the loss function $F(x)$. For *ImageNet* we use $\lambda = 10^{-5}$ and for *CIFAR* we use $\lambda = 10^{-4}$. We also compute the stochastic gradient using mini-batches of sample size 128.

## 5.2 Experimental results

For all experiments in this section we use *EASGD* with $\beta = 0.9$[8], for all momentum-based methods we set the momentum term $\delta = 0.99$ and finally for *MVADOWNPOUR* we set the moving rate to $\alpha = 0.001$. We start with the experiment on *CIFAR* dataset with $p = 4$ local workers running on a single computing node. For all the methods, we examined the communication periods from the following set $\tau = \{1, 4, 16, 64\}$. For comparison we also report the performance of *MSGD* which outperformed *SGD*, *ASGD* and *MVASGD* as shown in Figure 6 in the Supplement. For each method we examined a wide range of learning rates (the learning rates explored in all experiments are summarized in Table 1, 2, 3 in the Supplement). The *CIFAR* experiment was run 3 times independently from the same initialization and for each method we report its best performance measured by the smallest achievable test error. From the results in Figure 2, we conclude that all *DOWNPOUR*-based methods achieve their best performance (test error) for small $\tau$ ($\tau \in \{1, 4\}$), and become highly unstable for $\tau \in \{16, 64\}$. While *EAMSGD* significantly outperforms comparator methods for all values of $\tau$ by having faster convergence. It also finds better-quality solution measured by the test error and this advantage becomes more significant for $\tau \in \{16, 64\}$. Note that the tendency to achieve better test performance with larger $\tau$ is also characteristic for the *EASGD* algorithm.

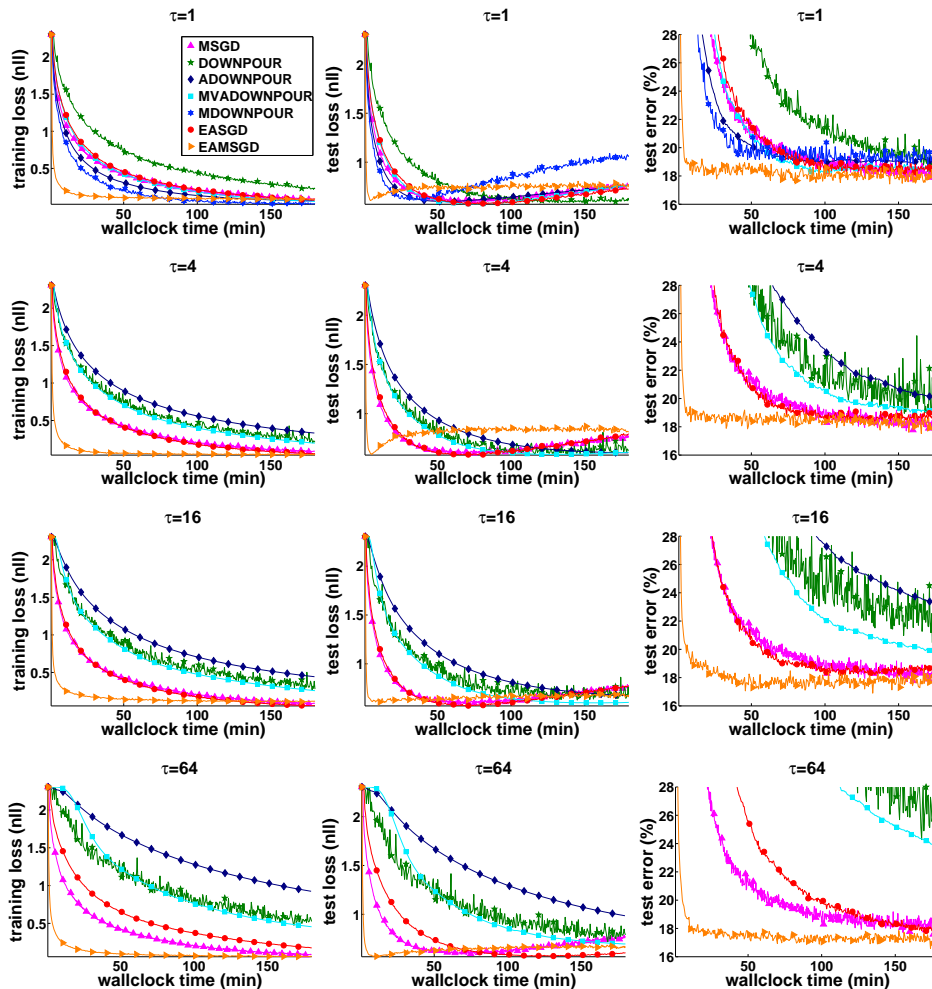

Figure 2: Training and test loss and the test error for the center variable versus a wallclock time for different communication periods $\tau$ on *CIFAR* dataset with the 7-layer convolutional neural network.

We next explore different number of local workers $p$ from the set $p = \{4, 8, 16\}$ for the *CIFAR* experiment, and $p = \{4, 8\}$ for the *ImageNet* experiment[9]. For the *ImageNet* experiment we report the results of one run with the best setting we have found. *EASGD* and *EAMSGD* were run with $\tau = 10$ whereas *DOWNPOUR* and *MDOWNPOUR* were run with $\tau = 1$. The results are in Figure 3 and 4. For the *CIFAR* experiment, it's noticeable that the lowest achievable test error by either *EASGD* or *EAMSGD* decreases with larger $p$. This can potentially be explained by the fact that larger $p$ allows for more exploration of the parameter space. In the Supplement, we discuss further the trade-off between exploration and exploitation as a function of the learning rate (section 9.5) and the communication period (section 9.6). Finally, the results obtained for the *ImageNet* experiment also shows the advantage of *EAMSGD* over the competitor methods.

## 6   Conclusion

In this paper we describe a new algorithm called *EASGD* and its variants for training deep neural networks in the stochastic setting when the computations are parallelized over multiple GPUs. Experiments demonstrate that this new algorithm quickly achieves improvement in test error compared to more common baseline approaches such as *DOWNPOUR* and its variants. We show that our approach is very stable and plausible under communication constraints. We provide the stability analysis of the asynchronous *EASGD* in the round-robin scheme, and show the theoretical advantage of the method over *ADMM*. The different behavior of the *EASGD* algorithm from its momentum-based variant *EAMSGD* is intriguing and will be studied in future works.

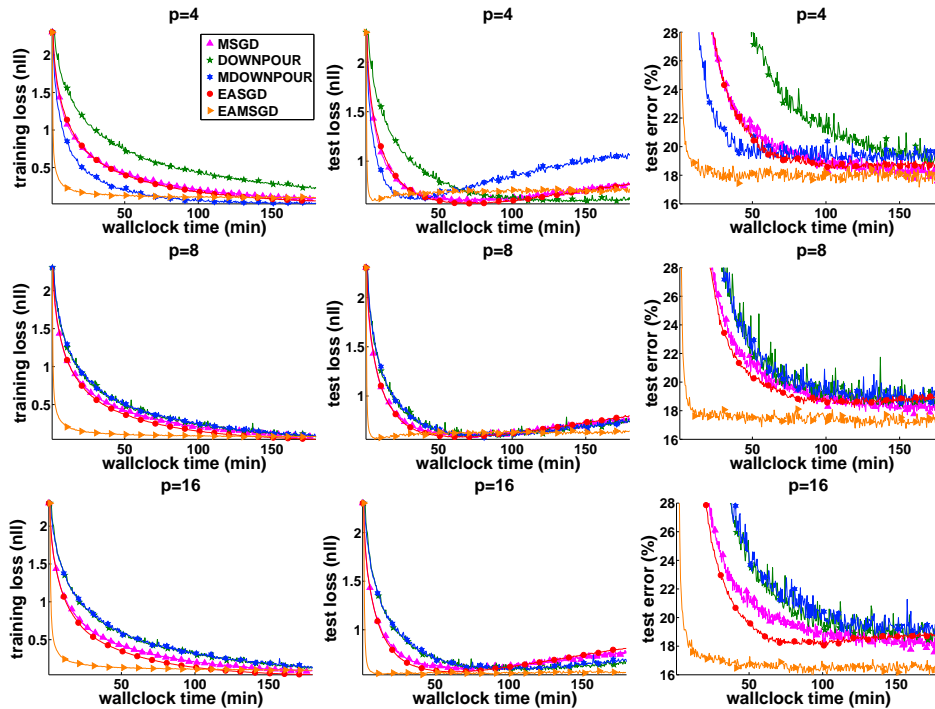

Figure 3: Training and test loss and the test error for the center variable versus a wallclock time for different number of local workers $p$ for parallel methods (*MSGD* uses $p = 1$) on *CIFAR* with the 7-layer convolutional neural network. *EAMSGD* achieves significant accelerations compared to other methods, e.g. the relative speed-up for $p = 16$ (the best comparator method is then *MSGD*) to achieve the test error $21\%$ equals $11.1$.

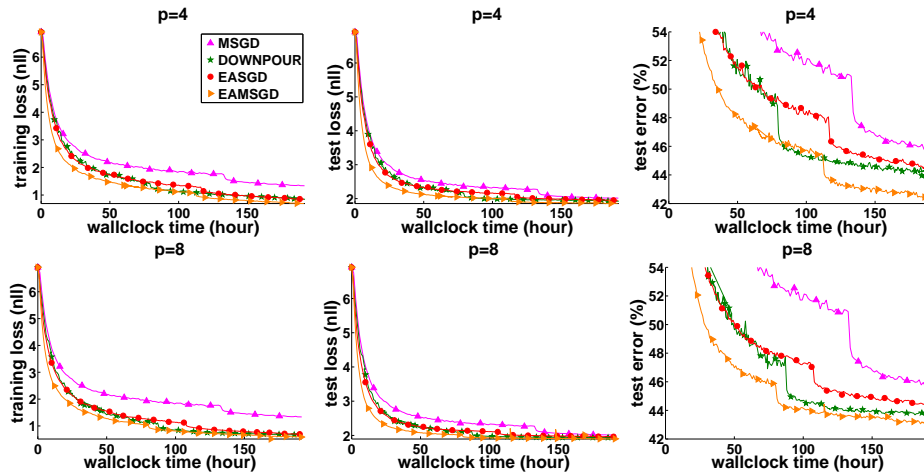

Figure 4: Training and test loss and the test error for the center variable versus a wallclock time for different number of local workers $p$ (*MSGD* uses $p = 1$) on *ImageNet* with the 11-layer convolutional neural network. Initial learning rate is decreased twice, by a factor of $5$ and then $2$, when we observe that the online predictive loss [30] stagnates. *EAMSGD* achieves significant accelerations compared to other methods, e.g. the relative speed-up for $p = 8$ (the best comparator method is then *DOWNPOUR*) to achieve the test error $49\%$ equals $1.8$, and simultaneously it reduces the communication overhead (*DOWNPOUR* uses communication period $\tau = 1$ and *EAMSGD* uses $\tau = 10$).

## Acknowledgments

The authors thank R. Power, J. Li for implementation guidance, J. Bruna, O. Henaff, C. Farabet, A. Szlam, Y. Bakhtin for helpful discussion, P. L. Combettes, S. Bengio and the referees for valuable feedback.

## Footnotes

[1]This condition resembles the stability condition for the synchronous *EASGD* algorithm (Condition 17 for $p = 1$) in the analysis in the Supplement.

[2]The convergence analysis in [27] is based on the assumption that "At any master iteration, updates from the workers have the same probability of arriving at the master.", which is not satisfied in the round-robin scheme.

[3]We have compared asynchronous *ADMM* [27] with *EASGD* in our setting as well, the performance is nearly the same. However, *ADMM*'s momentum variant is not as stable for large communication periods.

[4]Downloaded from `http://www.cs.toronto.edu/~kriz/cifar.html`.

[5]Downloaded from `http://image-net.org/challenges/LSVRC/2013`.

[6]Our implementation is available at `https://github.com/sixin-zh/mpiT`.

[7]On the contrary, initializing the local workers and the master with different random seeds 'traps' the algorithm in the symmetry breaking phase.

[8]Intuitively the 'effective $\beta$' is $\beta/\tau = p\alpha = p\eta\rho$ (thus $\rho = \frac{\beta}{\tau p \eta}$) in the asynchronous setting.

[9]For the *ImageNet* experiment, the training loss is measured on a subset of the training data of size 50,000.

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
