[Supplementary Material · easgd_nips2015_supp.pdf]

# Deep learning with Elastic Averaging SGD (Supplementary Material)

## 7 Additional theoretical results and proofs

### 7.1 Quadratic case

We provide here the convergence analysis of the synchronous *EASGD* algorithm with constant learning rate. The analysis is focused on the convergence of the center variable to the local optimum. We discuss one-dimensional quadratic case first, then the generalization to multi-dimensional setting (Lemma 7.3) and finally to the strongly convex case (Theorem 7.1).

Our analysis in the quadratic case extends the analysis of ASGD in [6]. Assume each of the $p$ local workers $x_t^i \in \mathbb{R}^n$ observes a noisy gradient at time $t \geq 0$ of the linear form given in Equation 14.

$$g_t^i(x_t^i) = Ax_t^i - b - \xi_t^i, \quad i \in \{1, \ldots, p\}, \tag{14}$$

where the matrix $A$ is positive-definite (each eigenvalue is strictly positive) and $\{\xi_t^i\}$'s are i.i.d. random variables, with zero mean and positive-definite covariance $\Sigma$. Let $x^*$ denote the optimum solution, where $x^* = A^{-1}b \in \mathbb{R}^n$. In this section we analyze the behavior of the mean squared error (MSE) of the center variable $\tilde{x}_t$, where this error is denoted as $\mathbb{E}[\|\tilde{x}_t - x^*\|^2]$, as a function of $t$, $p$, $\eta$, $\alpha$ and $\beta$, where $\beta = p\alpha$. Note that the MSE error can be decomposed as (squared) bias and variance[10]: $\mathbb{E}[\|\tilde{x}_t - x^*\|^2] = \|\mathbb{E}[\tilde{x}_t - x^*]\|^2 + \mathbb{V}[\tilde{x}_t - x^*]$. For one-dimensional case ($n = 1$), we assume $A = h > 0$ and $\Sigma = \sigma^2 > 0$.

**Lemma 7.1.** *Let $\tilde{x}_0$ and $\{x_0^i\}_{i=1,\ldots,p}$ be arbitrary constants, then*

$$\mathbb{E}[\tilde{x}_t - x^*] = \gamma^t(\tilde{x}_0 - x^*) + \frac{\gamma^t - \phi^t}{\gamma - \phi}\alpha u_0, \tag{15}$$

$$\mathbb{V}[\tilde{x}_t - x^*] = \frac{p^2\alpha^2\eta^2}{(\gamma - \phi)^2}\left(\frac{\gamma^2 - \gamma^{2t}}{1 - \gamma^2} + \frac{\phi^2 - \phi^{2t}}{1 - \phi^2} - 2\frac{\gamma\phi - (\gamma\phi)^t}{1 - \gamma\phi}\right)\frac{\sigma^2}{p}, \tag{16}$$

*where $u_0 = \sum_{i=1}^p (x_0^i - x^* - \frac{\alpha}{1 - p\alpha - \phi}(\tilde{x}_0 - x^*))$, $a = \eta h + (p+1)\alpha$, $c^2 = \eta h p\alpha$, $\gamma = 1 - \frac{a - \sqrt{a^2 - 4c^2}}{2}$, and $\phi = 1 - \frac{a + \sqrt{a^2 - 4c^2}}{2}$.*

It follows from Lemma 7.1 that for the center variable to be stable the following has to hold

$$-1 < \phi < \gamma < 1. \tag{17}$$

It can be verified that $\phi$ and $\gamma$ are the two zero-roots of the polynomial in $\lambda$: $\lambda^2 - (2 - a)\lambda + (1 - a + c^2)$. Recall that $\phi$ and $\lambda$ are the functions of $\eta$ and $\alpha$. Thus (see proof in Section 7.1.2)

- $\gamma < 1$ iff $c^2 > 0$ (i.e. $\eta > 0$ and $\alpha > 0$).
- $\phi > -1$ iff $(2 - \eta h)(2 - p\alpha) > 2\alpha$ and $(2 - \eta h) + (2 - p\alpha) > \alpha$.
- $\phi = \gamma$ iff $a^2 = 4c^2$ (i.e. $\eta h = \alpha = 0$).

The proof the above Lemma is based on the diagonalization of the linear gradient map (this map is symmetric due to the relation $\beta = p\alpha$). The stability analysis of the asynchronous *EASGD* algorithm in the round-robin scheme is similar due to this elastic symmetry.

*Proof.* Substituting the gradient from Equation 14 into the update rule used by each local worker in the synchronous *EASGD* algorithm (Equation 5 and 6) we obtain

$$x_{t+1}^i = x_t^i - \eta(Ax_t^i - b - \xi_t^i) - \alpha(x_t^i - \tilde{x}_t), \tag{18}$$

$$\tilde{x}_{t+1} = \tilde{x}_t + \sum_{i=1}^p \alpha(x_t^i - \tilde{x}_t), \tag{19}$$

where $\eta$ is the learning rate, and $\alpha$ is the moving rate. Recall that $\alpha = \eta\rho$ and $A = h$.

For the ease of notation we redefine $\tilde{x}_t$ and $x_t^i$ as follows:

$$\tilde{x}_t \triangleq \tilde{x}_t - x^* \quad \text{and} \quad x_t^i \triangleq x_t^i - x^*.$$

We prove the lemma by explicitly solving the linear equations 18 and 19. Let $x_t = (x_t^1, \ldots, x_t^p, \tilde{x}_t)^T$. We rewrite the recursive relation captured in Equation 18 and 19 as simply

$$x_{t+1} = Mx_t + b_t,$$

where the drift matrix $M$ is defined as

$$M = \begin{bmatrix} 1-\alpha-\eta h & 0 & \ldots & 0 & \alpha \\ 0 & 1-\alpha-\eta h & 0 & \ldots & \alpha \\ \ldots & 0 & \ldots & 0 & \ldots \\ 0 & \ldots & 0 & 1-\alpha-\eta h & \alpha \\ \alpha & \alpha & \ldots & \alpha & 1-p\alpha \end{bmatrix},$$

and the (diffusion) vector $b_t = (\eta\xi_t^1, \ldots, \eta\xi_t^p, 0)^T$.

Note that one of the eigenvalues of matrix $M$, that we call $\phi$, satisfies $(1-\alpha-\eta h-\phi)(1-p\alpha-\phi) = p\alpha^2$. The corresponding eigenvector is $(1, 1, \ldots, 1, -\frac{p\alpha}{1-p\alpha-\phi})^T$. Let $u_t$ be the projection of $x_t$ onto this eigenvector. Thus $u_t = \sum_{i=1}^p (x_t^i - \frac{\alpha}{1-p\alpha-\phi}\tilde{x}_t)$. Let furthermore $\xi_t = \sum_{i=1}^p \xi_t^i$. Therefore we have

$$u_{t+1} = \phi u_t + \eta\xi_t. \tag{20}$$

By combining Equation 19 and 20 as follows

$$\tilde{x}_{t+1} = \tilde{x}_t + \sum_{i=1}^p \alpha(x_t^i - \tilde{x}_t) = (1-p\alpha)\tilde{x}_t + \alpha(u_t + \frac{p\alpha}{1-p\alpha-\phi}\tilde{x}_t)$$

$$= (1-p\alpha + \frac{p\alpha^2}{1-p\alpha-\phi})\tilde{x}_t + \alpha u_t = \gamma\tilde{x}_t + \alpha u_t,$$

where the last step results from the following relations: $\frac{p\alpha^2}{1-p\alpha-\phi} = 1-\alpha-\eta h-\phi$ and $\phi + \gamma = 1-\alpha-\eta h + 1-p\alpha$. Thus we obtained

$$\tilde{x}_{t+1} = \gamma\tilde{x}_t + \alpha u_t. \tag{21}$$

Based on Equation 20 and 21, we can then expand $u_t$ and $\tilde{x}_t$ recursively,

$$u_{t+1} = \phi^{t+1}u_0 + \phi^t(\eta\xi_0) + \ldots + \phi^0(\eta\xi_t), \tag{22}$$

$$\tilde{x}_{t+1} = \gamma^{t+1}\tilde{x}_0 + \gamma^t(\alpha u_0) + \ldots + \gamma^0(\alpha u_t). \tag{23}$$

Substituting $u_0, u_1, \ldots, u_t$, each given through Equation 22, into Equation 23 we obtain

$$\tilde{x}_t = \gamma^t\tilde{x}_0 + \frac{\gamma^t - \phi^t}{\gamma - \phi}\alpha u_0 + \alpha\eta\sum_{l=1}^{t-1}\frac{\gamma^{t-l} - \phi^{t-l}}{\gamma - \phi}\xi_{l-1}. \tag{24}$$

To be more specific, the Equation 24 is obtained by integrating by parts,

$$\tilde{x}_{t+1} = \gamma^{t+1}\tilde{x}_0 + \sum_{i=0}^t \gamma^{t-i}(\alpha u_i)$$

$$= \gamma^{t+1}\tilde{x}_0 + \sum_{i=0}^t \gamma^{t-i}(\alpha(\phi^i u_0 + \sum_{l=0}^{i-1}\phi^{i-1-l}\eta\xi_l))$$

$$= \gamma^{t+1}\tilde{x}_0 + \sum_{i=0}^t \gamma^{t-i}\phi^i(\alpha u_0) + \sum_{l=0}^{t-1}\sum_{i=l+1}^t \gamma^{t-i}\phi^{i-1-l}(\alpha\eta\xi_l)$$

$$= \gamma^{t+1}\tilde{x}_0 + \frac{\gamma^{t+1} - \phi^{t+1}}{\gamma - \phi}(\alpha u_0) + \sum_{l=0}^{t-1}\frac{\gamma^{t-l} - \phi^{t-l}}{\gamma - \phi}(\alpha\eta\xi_l).$$

Since the random variables $\xi_l$ are i.i.d, we may sum the variance term by term as follows

$$\sum_{l=0}^{t-1}\left(\frac{\gamma^{t-l}-\phi^{t-l}}{\gamma-\phi}\right)^2 = \sum_{l=0}^{t-1}\frac{\gamma^{2(t-l)}-2\gamma^{t-l}\phi^{t-l}+\phi^{2(t-l)}}{(\gamma-\phi)^2}$$

$$= \frac{1}{(\gamma-\phi)^2}\left(\frac{\gamma^2-\gamma^{2(t+1)}}{1-\gamma^2}-2\frac{\gamma\phi-(\gamma\phi)^{t+1}}{1-\gamma\phi}+\frac{\phi^2-\phi^{2(t+1)}}{1-\phi^2}\right). \quad (25)$$

Note that $\mathbb{E}[\xi_t] = \sum_{i=1}^{p}\mathbb{E}[\xi_t^i] = 0$ and $\mathbb{V}[\xi_t] = \sum_{i=1}^{p}\mathbb{V}[\xi_t^i] = p\sigma^2$. These two facts, the equality in Equation 24 and Equation 25 can then be used to compute $\mathbb{E}[\tilde{x}_t]$ and $\mathbb{V}[\tilde{x}_t]$ as given in Equation 15 and 16 in Lemma 7.1. $\qquad\square$

### 7.1.1 Visualizing Lemma 7.1

Figure 5: Theoretical mean squared error (MSE) of the center $\tilde{x}$ in the quadratic case, with various choices of the learning rate $\eta$ (horizontal within each block), and the moving rate $\beta = p\alpha$ (vertical within each block), the number of processors $p = \{1, 10, 100, 1000, 10000\}$ (vertical across blocks), and the time steps $t = \{1, 2, 10, 100, \infty\}$ (horizontal across blocks). The MSE is plotted in log scale, ranging from $10^{-3}$ to $10^3$ (from deep blue to red). The dark red (i.e. on the upper-right corners) indicates divergence.

In Figure 5, we illustrate the dependence of MSE on $\beta$, $\eta$ and the number of processors $p$ over time $t$. We consider the large-noise setting where $\tilde{x}_0 = x_0^i = 1$, $h = 1$ and $\sigma = 10$. The MSE error is color-coded such that the deep blue color corresponds to the MSE equal to $10^{-3}$, the green color corresponds to the MSE equal to $1$, the red color corresponds to MSE equal to $10^3$ and the dark red color corresponds to the divergence of algorithm *EASGD* (condition in Equation 17 is then violated). The plot shows that we can achieve significant variance reduction by increasing the number of local workers $p$. This effect is less sensitive to the choice of $\beta$ and $\eta$ for large $p$.

### 7.1.2 Condition in Equation 17

We are going to show that

- $\gamma < 1$ iff $c^2 > 0$ (i.e. $\eta > 0$ and $\beta > 0$).
- $\phi > -1$ iff $(2 - \eta h)(2 - \beta) > 2\beta/p$ and $(2 - \eta h) + (2 - \beta) > \beta/p$.
- $\phi = \gamma$ iff $a^2 = 4c^2$ (i.e. $\eta h = \beta = 0$).

Recall that $a = \eta h + (p+1)\alpha$, $c^2 = \eta h p \alpha$, $\gamma = 1 - \frac{a - \sqrt{a^2 - 4c^2}}{2}$, $\phi = 1 - \frac{a + \sqrt{a^2 - 4c^2}}{2}$, and $\beta = p\alpha$. We have

- $\gamma < 1 \Leftrightarrow \frac{a - \sqrt{a^2 - 4c^2}}{2} > 0 \Leftrightarrow a > \sqrt{a^2 - 4c^2} \Leftrightarrow a^2 > a^2 - 4c^2 \Leftrightarrow c^2 > 0$.
- $\phi > -1 \Leftrightarrow 2 > \frac{a + \sqrt{a^2 - 4c^2}}{2} \Leftrightarrow 4 - a > \sqrt{a^2 - 4c^2} \Leftrightarrow 4 - a > 0, (4 - a)^2 > a^2 - 4c^2 \Leftrightarrow$
  $4 - a > 0, 4 - 2a + c^2 > 0 \Leftrightarrow 4 > \eta h + \beta + \alpha, 4 - 2(\eta h + \beta + \alpha) + \eta h \beta > 0$.
- $\phi = \gamma \Leftrightarrow \sqrt{a^2 - 4c^2} = 0 \Leftrightarrow a^2 = 4c^2$.

The next corollary is a consequence of Lemma 7.1. As the number of workers $p$ grows, the averaging property of the *EASGD* can be characterized as follows

**Corollary 7.1.** *Let the Elastic Averaging relation $\beta = p\alpha$ and the condition 17 hold, then*

$$\lim_{p \to \infty} \lim_{t \to \infty} p \mathbb{E}[(\tilde{x}_t - x^*)^2] = \frac{\beta \eta h}{(2 - \beta)(2 - \eta h)} \cdot \frac{2 - \beta - \eta h + \beta \eta h}{\beta + \eta h - \beta \eta h} \cdot \frac{\sigma^2}{h^2}.$$

*Proof.* Note that when $\beta$ is fixed, $\lim_{p \to \infty} a = \eta h + \beta$ and $c^2 = \eta h \beta$. Then $\lim_{p \to \infty} \phi = \min(1 - \beta, 1 - \eta h)$ and $\lim_{p \to \infty} \gamma = \max(1 - \beta, 1 - \eta h)$. Also note that using Lemma 7.1 we obtain

$$\lim_{t \to \infty} \mathbb{E}[(\tilde{x}_t - x^*)^2] = \frac{\beta^2 \eta^2}{(\gamma - \phi)^2} \left( \frac{\gamma^2}{1 - \gamma^2} + \frac{\phi^2}{1 - \phi^2} - \frac{2\gamma\phi}{1 - \gamma\phi} \right) \frac{\sigma^2}{p}$$

$$= \frac{\beta^2 \eta^2}{(\gamma - \phi)^2} \left( \frac{\gamma^2(1 - \phi^2)(1 - \phi\gamma) + \phi^2(1 - \gamma^2)(1 - \phi\gamma) - 2\gamma\phi(1 - \gamma^2)(1 - \phi^2)}{(1 - \gamma^2)(1 - \phi^2)(1 - \gamma\phi)} \right) \frac{\sigma^2}{p}$$

$$= \frac{\beta^2 \eta^2}{(\gamma - \phi)^2} \left( \frac{(\gamma - \phi)^2(1 + \gamma\phi)}{(1 - \gamma^2)(1 - \phi^2)(1 - \gamma\phi)} \right) \frac{\sigma^2}{p}$$

$$= \frac{\beta^2 \eta^2}{(1 - \gamma^2)(1 - \phi^2)} \cdot \frac{1 + \gamma\phi}{1 - \gamma\phi} \cdot \frac{\sigma^2}{p}.$$

Corollary 7.1 is obtained by pluging in the limiting values of $\phi$ and $\gamma$. $\qquad \square$

The crucial point of Corollary 7.1 is that the MSE in the limit $t \to \infty$ is in the order of $1/p$ which implies that as the number of processors $p$ grows, the MSE will decrease for the *EASGD* algorithm. Also note that the smaller the $\beta$ is (recall that $\beta = p\alpha = p\eta\rho$), the more exploration is allowed (small $\rho$) and simultaneously the smaller the MSE is.

### 7.2 Generalization to multidimensional case

The next lemma (Lemma 7.2) shows that *EASGD* algorithm achieves the highest possible rate of convergence when we consider the double averaging sequence (similarly to [6]) $\{z_1, z_2, \dots\}$ defined as below

$$z_{t+1} = \frac{1}{t + 1} \sum_{k=0}^{t} \tilde{x}_k. \tag{26}$$

**Lemma 7.2** (Weak convergence)**.** *If the condition in Equation 17 holds, then the normalized double averaging sequence defined in Equation 26 converges weakly to the normal distribution with zero mean and variance $\sigma^2/ph^2$,*

$$\sqrt{t}(z_t - x^*) \rightharpoonup \mathcal{N}(0, \frac{\sigma^2}{ph^2}), \quad t \to \infty. \tag{27}$$

*Proof.* As in the proof of Lemma 7.1, for the ease of notation we redefine $\tilde{x}_t$ and $x_t^i$ as follows:

$$\tilde{x}_t \triangleq \tilde{x}_t - x^* \quad \text{and} \quad x_t^i \triangleq x_t^i - x^*.$$

Also recall that $\{\xi_t^i\}$'s are *i.i.d.* random variables (noise) with zero mean and the same covariance $\Sigma \succ 0$. We are interested in the asymptotic behavior of the double averaging sequence $\{z_1, z_2, \dots\}$ defined as

$$z_{t+1} = \frac{1}{t+1} \sum_{k=0}^{t} \tilde{x}_k. \tag{28}$$

Recall the Equation 24 from the proof of Lemma 7.1 (for the convenience it is provided below):

$$\tilde{x}_k = \gamma^k \tilde{x}_0 + \alpha u_0 \frac{\gamma^k - \phi^k}{\gamma - \phi} + \alpha \eta \sum_{l=1}^{k-1} \frac{\gamma^{k-l} - \phi^{k-l}}{\gamma - \phi} \xi_{l-1},$$

where $\xi_t = \sum_{i=1}^{p} \xi_t^i$. Therefore

$$\sum_{k=0}^{t} \tilde{x}_k = \frac{1-\gamma^{t+1}}{1-\gamma} \tilde{x}_0 + \alpha u_0 \frac{1}{\gamma - \mu} \left( \frac{1-\gamma^{t+1}}{1-\gamma} - \frac{1-\phi^{t+1}}{1-\phi} \right) + \alpha \eta \sum_{l=1}^{t-1} \sum_{k=l+1}^{t} \frac{\gamma^{k-l} - \phi^{k-l}}{\gamma - \phi} \xi_{l-1}$$

$$= O(1) + \alpha \eta \sum_{l=1}^{t-1} \frac{1}{\gamma - \phi} \left( \gamma \frac{1-\gamma^{t-l}}{1-\gamma} - \phi \frac{1-\phi^{t-l}}{1-\phi} \right) \xi_{l-1}$$

Note that the only non-vanishing term (in weak convergence) of $1/\sqrt{t} \sum_{k=0}^{t} \tilde{x}_k$ as $t \to \infty$ is

$$\frac{1}{\sqrt{t}} \alpha \eta \sum_{l=1}^{t-1} \frac{1}{\gamma - \phi} \left( \frac{\gamma}{1-\gamma} - \frac{\phi}{1-\phi} \right) \xi_{l-1}. \tag{29}$$

Also recall that $\mathbb{V}[\xi_{l-1}] = p\sigma^2$ and

$$\frac{1}{\gamma - \phi} \left( \frac{\gamma}{1-\gamma} - \frac{\phi}{1-\phi} \right) = \frac{1}{(1-\gamma)(1-\phi)} = \frac{1}{\eta h p \alpha}.$$

Therefore the expression in Equation 29 is asymptotically normal with zero mean and variance $\sigma^2/ph^2$. $\qquad\square$

The asymptotic variance in the Lemma 7.2 is optimal with any fixed $\eta$ and $\beta$ for which Equation 17 holds. The next lemma (Lemma 7.3) extends the result in Lemma 7.2 to the multi-dimensional setting.

**Lemma 7.3** (Weak convergence). *Let $h$ denotes the largest eigenvalue of $A$. If $(2 - \eta h)(2 - \beta) > 2\beta/p$, $(2 - \eta h) + (2 - \beta) > \beta/p$, $\eta > 0$ and $\beta > 0$, then the normalized double averaging sequence converges weakly to the normal distribution with zero mean and the covariance matrix $V = A^{-1}\Sigma(A^{-1})^T$,*

$$\sqrt{tp}(z_t - x^*) \rightharpoonup \mathcal{N}(0, V), \quad t \to \infty. \tag{30}$$

*Proof.* Since $A$ is symmetric, one can use the proof technique of Lemma 7.2 to prove Lemma 7.3 by diagonalizing the matrix $A$. This diagonalization essentially generalizes Lemma 7.1 to the multidimensional case. We will not go into the details of this proof as we will provide a simpler way to look at the system. As in the proof of Lemma 7.1 and Lemma 7.2, for the ease of notation we redefine $\tilde{x}_t$ and $x_t^i$ as follows:

$$\tilde{x}_t \triangleq \tilde{x}_t - x^* \quad \text{and} \quad x_t^i \triangleq x_t^i - x^*.$$

Let the spatial average of the local parameters at time $t$ be denoted as $y_t$ where $y_t = \frac{1}{p} \sum_{i=1}^{p} x_t^i$, and let the average noise be denoted as $\xi_t$, where $\xi_t = \frac{1}{p} \sum_{i=1}^{p} \xi_t^i$. Equations 18 and 19 can then be reduced to the following

$$y_{t+1} = y_t - \eta(Ay_t - \xi_t) + \alpha(\tilde{x}_t - y_t), \tag{31}$$
$$\tilde{x}_{t+1} = \tilde{x}_t + \beta(y_t - \tilde{x}_t). \tag{32}$$

We focus on the case where the learning rate $\eta$ and the moving rate $\alpha$ are kept constant over time[11]. Recall $\beta = p\alpha$ and $\alpha = \eta\rho$.

Let's introduce the block notation $U_t = (y_t, \tilde{x}_t)$, $\Xi_t = (\eta\xi_t, 0)$, $M = I - \eta L$ and

$$L = \begin{pmatrix} A + \frac{\alpha}{\eta}I & -\frac{\alpha}{\eta}I \\ -\frac{\beta}{\eta}I & \frac{\beta}{\eta}I \end{pmatrix}.$$

From Equations 31 and 32 it follows that $U_{t+1} = MU_t + \Xi_t$. Note that this linear system has a degenerate noise $\Xi_t$ which prevents us from directly applying results of [6]. Expanding this recursive relation and summing by parts, we have

$$\begin{aligned} \sum_{k=0}^{t} U_k \quad = \quad & M^0 U_0 + \\ & M^1 U_0 + M^0\Xi_0 + \\ & M^2 U_0 + M^1\Xi_0 + M^0\Xi_1 + \\ & \dots \\ & M^t U_0 + M^{t-1}\Xi_0 + \dots + M^0\Xi_{t-1}. \end{aligned}$$

By Lemma 7.4, $\|M\|_2 < 1$ and thus

$$M^0 + M^1 + \dots + M^t + \dots = (I - M)^{-1} = \eta^{-1}L^{-1}.$$

Since $A$ is invertible, we get

$$L^{-1} = \begin{pmatrix} A^{-1} & \frac{\alpha}{\beta}A^{-1} \\ A^{-1} & \frac{\eta}{\beta} + \frac{\alpha}{\beta}A^{-1} \end{pmatrix},$$

thus

$$\frac{1}{\sqrt{t}}\sum_{k=0}^{t} U_k = \frac{1}{\sqrt{t}}U_0 + \frac{1}{\sqrt{t}}\eta L^{-1}\sum_{k=1}^{t}\Xi_{k-1} - \frac{1}{\sqrt{t}}\sum_{k=1}^{t}M^{k+1}\Xi_{k-1}.$$

Note that the only non-vanishing term (in weak convergence) of $\frac{1}{\sqrt{t}}\sum_{k=0}^{t} U_k$ is $\frac{1}{\sqrt{t}}(\eta L)^{-1}\sum_{k=1}^{t}\Xi_{k-1}$ thus we have

$$\frac{1}{\sqrt{t}}(\eta L)^{-1}\sum_{k=1}^{t}\Xi_{k-1} \rightharpoonup \mathcal{N}\left(\begin{pmatrix} 0 \\ 0 \end{pmatrix}, \begin{pmatrix} V & V \\ V & V \end{pmatrix}\right), \tag{33}$$

where $V = A^{-1}\Sigma(A^{-1})^T$. □

**Lemma 7.4.** *If the following conditions hold:*

$$\begin{aligned} (2 - \eta h)(2 - p\alpha) &> 2\alpha \\ (2 - \eta h) + (2 - p\alpha) &> \alpha \\ \eta &> 0 \\ \alpha &> 0 \end{aligned}$$

*then* $\|M\|_2 < 1$.

*Proof.* The eigenvalue $\lambda$ of $M$ and the (non-zero) eigenvector $(y, z)$ of $M$ satisfy

$$M\begin{pmatrix} y \\ z \end{pmatrix} = \lambda\begin{pmatrix} y \\ z \end{pmatrix}. \tag{34}$$

Recall that

$$M = I - \eta L = \begin{pmatrix} I - \eta A - \alpha I & \alpha I \\ \beta I & I - \beta I \end{pmatrix}. \tag{35}$$

From the Equations 34 and 35 we obtain

$$\begin{cases} y - \eta A y - \alpha y + \alpha z = \lambda y \\ \beta y + (1 - \beta) z = \lambda z \end{cases}. \tag{36}$$

Since $(y, z)$ is assumed to be non-zero, we can write $z = \beta y / (\lambda + \beta - 1)$. Then the Equation 36 can be reduced to

$$\eta A y = (1 - \alpha - \lambda) y + \frac{\alpha \beta}{\lambda + \beta - 1} y. \tag{37}$$

Thus $y$ is the eigenvector of $A$. Let $\lambda_A$ be the eigenvalue of matrix $A$ such that $Ay = \lambda_A y$. Thus based on Equation 37 it follows that

$$\eta \lambda_A = (1 - \alpha - \lambda) + \frac{\alpha \beta}{\lambda + \beta - 1}. \tag{38}$$

Equation 38 is equivalent to

$$\lambda^2 - (2 - a) \lambda + (1 - a + c^2) = 0, \tag{39}$$

where $a = \eta \lambda_A + (p + 1) \alpha$, $c^2 = \eta \lambda_A p \alpha$. It follows from the condition in Equation 17 that $-1 < \lambda < 1$ iff $\eta > 0$, $\beta > 0$, $(2 - \eta \lambda_A)(2 - \beta) > 2\beta/p$ and $(2 - \eta \lambda_A) + (2 - \beta) > \beta/p$. Let $h$ denote the maximum eigenvalue of $A$ and note that $2 - \eta \lambda_A \geq 2 - \eta h$. This implies that the condition of our lemma is sufficient. $\square$

As in Lemma 7.2, the asymptotic covariance in the Lemma 7.3 is optimal, i.e. meets the Fisher information lower-bound. The fact that this asymptotic covariance matrix $V$ does not contain any term involving $\rho$ is quite remarkable, since the penalty term $\rho$ does have an impact on the condition number of the Hessian in Equation 2.

## 7.3  Strongly convex case

We now extend the above proof ideas to analyze the strongly convex case, in which the noisy gradient $g_t^i(x) = \nabla F(x) - \xi_t^i$ has the regularity that there exists some $0 < \mu \leq L$, for which $\mu \|x - y\|^2 \leq \langle \nabla F(x) - \nabla F(y), x - y \rangle \leq L \|x - y\|^2$ holds uniformly for any $x \in \mathbb{R}^d, y \in \mathbb{R}^d$. The noise $\{\xi_t^i\}$'s is assumed to be i.i.d. with zero mean and bounded variance $\mathbb{E}[\|\xi_t^i\|^2] \leq \sigma^2$.

**Theorem 7.1.** *Let* $a_t = \mathbb{E} \left\| \frac{1}{p} \sum_{i=1}^{p} x_t^i - x^* \right\|^2$, $b_t = \frac{1}{p} \sum_{i=1}^{p} \mathbb{E} \|x_t^i - x^*\|^2$, $c_t = \mathbb{E} \|\tilde{x}_t - x^*\|^2$, $\gamma_1 = 2\eta \frac{\mu L}{\mu + L}$ *and* $\gamma_2 = 2\eta L (1 - \frac{2\sqrt{\mu L}}{\mu + L})$. *If* $0 \leq \eta \leq \frac{2}{\mu + L}(1 - \alpha)$, $0 \leq \alpha < 1$ *and* $0 \leq \beta \leq 1$ *then*

$$\begin{pmatrix} a_{t+1} \\ b_{t+1} \\ c_{t+1} \end{pmatrix} \leq \begin{pmatrix} 1 - \gamma_1 - \gamma_2 - \alpha & \gamma_2 & \alpha \\ 0 & 1 - \gamma_1 - \alpha & \alpha \\ \beta & 0 & 1 - \beta \end{pmatrix} \begin{pmatrix} a_t \\ b_t \\ c_t \end{pmatrix} + \begin{pmatrix} \eta^2 \frac{\sigma^2}{p} \\ \eta^2 \sigma^2 \\ 0 \end{pmatrix}.$$

*Proof.* The idea of the proof is based on the point of view in Lemma 7.3, i.e. how close the center variable $\tilde{x}_t$ is to the spatial average of the local variables $y_t = \frac{1}{p} \sum_{i=1}^{p} x_t^i$. To further simplify the notation, let the noisy gradient be $\nabla f_{t,\xi}^i = g_t^i(x_t^i) = \nabla F(x_t^i) - \xi_t^i$, and $\nabla f_t^i = \nabla F(x_t^i)$ be its deterministic part. Then *EASGD* updates can be rewritten as follows,

$$x_{t+1}^i = x_t^i - \eta \nabla f_{t,\xi}^i - \alpha(x_t^i - \tilde{x}_t), \tag{40}$$

$$\tilde{x}_{t+1} = \tilde{x}_t + \beta(y_t - \tilde{x}_t). \tag{41}$$

We have thus the update for the spatial average,

$$y_{t+1} = y_t - \eta \frac{1}{p} \sum_{i=1}^{p} \nabla f_{t,\xi}^i - \alpha(y_t - \tilde{x}_t). \tag{42}$$

The idea of the proof is to bound the distance $\|\tilde{x}_t - x^*\|^2$ through $\|y_t - x^*\|^2$ and $\frac{1}{p}\sum_i^p \|x_t^i - x^*\|^2$. W start from the following estimate for the strongly convex function [31],

$$\langle \nabla F(x) - \nabla F(y), x - y \rangle \geq \frac{\mu L}{\mu + L} \|x - y\|^2 + \frac{1}{\mu + L} \|\nabla F(x) - \nabla F(y)\|^2 .$$

Since $\nabla f(x^*) = 0$, we have

$$\langle \nabla f_t^i, x_t^i - x^* \rangle \geq \frac{\mu L}{\mu + L} \|x_t^i - x^*\|^2 + \frac{1}{\mu + L} \|\nabla f_t^i\|^2 . \tag{43}$$

From Equation 40 the following relation holds,

$$
\begin{aligned}
\|x_{t+1}^i - x^*\|^2 &= \|x_t^i - x^*\|^2 + \eta^2 \|\nabla f_{t,\xi}^i\|^2 + \alpha^2 \|x_t^i - \tilde{x}_t\|^2 \\
&- 2\eta \langle \nabla f_{t,\xi}^i, x_t^i - x^* \rangle - 2\alpha \langle x_t^i - \tilde{x}_t, x_t^i - x^* \rangle \\
&+ 2\eta\alpha \langle \nabla f_{t,\xi}^i, x_t^i - \tilde{x}_t \rangle .
\end{aligned}
\tag{44}
$$

By the cosine rule ($2\langle a - b, c - d \rangle = \|a - d\|^2 - \|a - c\|^2 + \|c - b\|^2 - \|d - b\|^2$), we have

$$2\langle x_t^i - \tilde{x}_t, x_t^i - x^* \rangle = \|x_t^i - x^*\|^2 + \|x_t^i - \tilde{x}_t\|^2 - \|\tilde{x}_t - x^*\|^2 . \tag{45}$$

By the Cauchy-Schwarz inequality, we have

$$\langle \nabla f_t^i, x_t^i - \tilde{x}_t \rangle \leq \|\nabla f_t^i\| \|x_t^i - \tilde{x}_t\| . \tag{46}$$

Combining the above estimates in Equations 43, 44, 45, 46, we obtain

$$
\begin{aligned}
\|x_{t+1}^i - x^*\|^2 &\leq \|x_t^i - x^*\|^2 + \eta^2 \|\nabla f_t^i - \xi_t^i\|^2 + \alpha^2 \|x_t^i - \tilde{x}_t\|^2 \\
&- 2\eta \left( \frac{\mu L}{\mu + L} \|x_t^i - x^*\|^2 + \frac{1}{\mu + L} \|\nabla f_t^i\|^2 \right) + 2\eta \langle \xi_t^i, x_t^i - x^* \rangle \\
&- \alpha \left( \|x_t^i - x^*\|^2 + \|x_t^i - \tilde{x}_t\|^2 - \|\tilde{x}_t - x^*\|^2 \right) \\
&+ 2\eta\alpha \|\nabla f_t^i\| \|x_t^i - \tilde{x}_t\| - 2\eta\alpha \langle \xi_t^i, x_t^i - \tilde{x}_t \rangle .
\end{aligned}
\tag{47}
$$

Choosing $0 \leq \alpha < 1$, we can have this upper-bound for the terms $\alpha^2 \|x_t^i - \tilde{x}_t\|^2 - \alpha \|x_t^i - \tilde{x}_t\|^2 + 2\eta\alpha \|\nabla f_t^i\| \|x_t^i - \tilde{x}_t\| = -\alpha(1 - \alpha) \|x_t^i - \tilde{x}_t\|^2 + 2\eta\alpha \|\nabla f_t^i\| \|x_t^i - \tilde{x}_t\| \leq \frac{\eta^2 \alpha}{1 - \alpha} \|\nabla f_t^i\|^2$ by applying $-ax^2 + bx \leq \frac{b^2}{4a}$ with $x = \|x_t^i - \tilde{x}_t\|$. Thus we can further bound Equation 47 with

$$
\begin{aligned}
\|x_{t+1}^i - x^*\|^2 &\leq (1 - 2\eta \frac{\mu L}{\mu + L} - \alpha) \|x_t^i - x^*\|^2 + (\eta^2 + \frac{\eta^2 \alpha}{1 - \alpha} - \frac{2\eta}{\mu + L}) \|\nabla f_t^i\|^2 \\
&- 2\eta^2 \langle \nabla f_t^i, \xi_t^i \rangle + 2\eta \langle \xi_t^i, x_t^i - x^* \rangle - 2\eta\alpha \langle \xi_t^i, x_t^i - \tilde{x}_t \rangle \tag{48} \\
&+ \eta^2 \|\xi_t^i\|^2 + \alpha \|\tilde{x}_t - x^*\|^2 \tag{49}
\end{aligned}
$$

As in Equation 48 and 49, the noise $\xi_t^i$ is zero mean ($\mathbb{E}\xi_t^i = 0$) and the variance of the noise $\xi_t^i$ is bounded ($\mathbb{E}\|\xi_t^i\|^2 \leq \sigma^2$), if $\eta$ is chosen small enough such that $\eta^2 + \frac{\eta^2 \alpha}{1 - \alpha} - \frac{2\eta}{\mu + L} \leq 0$, then

$$\mathbb{E}\|x_{t+1}^i - x^*\|^2 \leq (1 - 2\eta \frac{\mu L}{\mu + L} - \alpha)\mathbb{E}\|x_t^i - x^*\|^2 + \eta^2 \sigma^2 + \alpha \mathbb{E}\|\tilde{x}_t - x^*\|^2 . \tag{50}$$

Now we apply similar idea to estimate $\|y_t - x^*\|^2$. From Equation 42 the following relation holds,

$$
\begin{aligned}
\|y_{t+1} - x^*\|^2 &= \|y_t - x^*\|^2 + \eta^2 \left\| \frac{1}{p}\sum_{i=1}^p \nabla f_{t,\xi}^i \right\|^2 + \alpha^2 \|y_t - \tilde{x}_t\|^2 \\
&- 2\eta \left\langle \frac{1}{p}\sum_{i=1}^p \nabla f_{t,\xi}^i, y_t - x^* \right\rangle - 2\alpha \langle y_t - \tilde{x}_t, y_t - x^* \rangle \\
&+ 2\eta\alpha \left\langle \frac{1}{p}\sum_{i=1}^p \nabla f_{t,\xi}^i, y_t - \tilde{x}_t \right\rangle .
\end{aligned}
\tag{51}
$$

By $\left\langle \frac{1}{p}\sum_{i=1}^{p} a_i, \frac{1}{p}\sum_{j=1}^{p} b_j \right\rangle = \frac{1}{p}\sum_{i=1}^{p}\langle a_i, b_i\rangle - \frac{1}{p^2}\sum_{i>j}\langle a_i - a_j, b_i - b_j\rangle$, we have

$$\left\langle \frac{1}{p}\sum_{i=1}^{p}\nabla f_t^i, y_t - x^* \right\rangle = \frac{1}{p}\sum_{i=1}^{p}\langle \nabla f_t^i, x_t^i - x^*\rangle - \frac{1}{p^2}\sum_{i>j}\left\langle \nabla f_t^i - \nabla f_t^j, x_t^i - x_t^j \right\rangle. \tag{52}$$

By the cosine rule, we have

$$2\langle y_t - \tilde{x}_t, y_t - x^*\rangle = \|y_t - x^*\|^2 + \|y_t - \tilde{x}_t\|^2 - \|\tilde{x}_t - x^*\|^2. \tag{53}$$

Denote $\xi_t = \frac{1}{p}\sum_{i=1}^{p}\xi_t^i$, we can rewrite Equation 51 as

$$\begin{aligned}
\|y_{t+1} - x^*\|^2 &= \|y_t - x^*\|^2 + \eta^2\left\|\frac{1}{p}\sum_{i=1}^{p}\nabla f_t^i - \xi_t\right\|^2 + \alpha^2\|y_t - \tilde{x}_t\|^2 \\
&\quad - 2\eta\left\langle\frac{1}{p}\sum_{i=1}^{p}\nabla f_t^i - \xi_t, y_t - x^*\right\rangle - 2\alpha\langle y_t - \tilde{x}_t, y_t - x^*\rangle \\
&\quad + 2\eta\alpha\left\langle\frac{1}{p}\sum_{i=1}^{p}\nabla f_t^i - \xi_t, y_t - \tilde{x}_t\right\rangle.
\end{aligned} \tag{54}$$

By combining the above Equations 52, 53 with 54, we obtain

$$\begin{aligned}
\|y_{t+1} - x^*\|^2 &= \|y_t - x^*\|^2 + \eta^2\left\|\frac{1}{p}\sum_{i=1}^{p}\nabla f_t^i - \xi_t\right\|^2 + \alpha^2\|y_t - \tilde{x}_t\|^2 \\
&\quad - 2\eta\left(\frac{1}{p}\sum_{i=1}^{p}\langle\nabla f_t^i, x_t^i - x^*\rangle - \frac{1}{p^2}\sum_{i>j}\left\langle\nabla f_t^i - \nabla f_t^j, x_t^i - x_t^j\right\rangle\right) \tag{55} \\
&\quad + 2\eta\langle\xi_t, y_t - x^*\rangle - \alpha(\|y_t - x^*\|^2 + \|y_t - \tilde{x}_t\|^2 - \|\tilde{x}_t - x^*\|^2) \\
&\quad + 2\eta\alpha\left\langle\frac{1}{p}\sum_{i=1}^{p}\nabla f_t^i - \xi_t, y_t - \tilde{x}_t\right\rangle.
\end{aligned} \tag{56}$$

Thus it follows from Equation 43 and 56 that

$$\begin{aligned}
\|y_{t+1} - x^*\|^2 &\leq \|y_t - x^*\|^2 + \eta^2\left\|\frac{1}{p}\sum_{i=1}^{p}\nabla f_t^i - \xi_t\right\|^2 + \alpha^2\|y_t - \tilde{x}_t\|^2 \\
&\quad - 2\eta\frac{1}{p}\sum_{i=1}^{p}\left(\frac{\mu L}{\mu + L}\|x_t^i - x^*\|^2 + \frac{1}{\mu + L}\|\nabla f_t^i\|^2\right) \\
&\quad + 2\eta\frac{1}{p^2}\sum_{i>j}\left\langle\nabla f_t^i - \nabla f_t^j, x_t^i - x_t^j\right\rangle \\
&\quad + 2\eta\langle\xi_t, y_t - x^*\rangle - \alpha(\|y_t - x^*\|^2 + \|y_t - \tilde{x}_t\|^2 - \|\tilde{x}_t - x^*\|^2) \\
&\quad + 2\eta\alpha\left\langle\frac{1}{p}\sum_{i=1}^{p}\nabla f_t^i - \xi_t, y_t - \tilde{x}_t\right\rangle.
\end{aligned} \tag{57}$$

Recall $y_t = \frac{1}{p}\sum_{i=1}^{p}x_t^i$, we have the following bias-variance relation,

$$\begin{aligned}
\frac{1}{p}\sum_{i=1}^{p}\|x_t^i - x^*\|^2 &= \frac{1}{p}\sum_{i=1}^{p}\|x_t^i - y_t\|^2 + \|y_t - x^*\|^2 = \frac{1}{p^2}\sum_{i>j}\left\|x_t^i - x_t^j\right\|^2 + \|y_t - x^*\|^2, \\
\frac{1}{p}\sum_{i=1}^{p}\|\nabla f_t^i\|^2 &= \frac{1}{p^2}\sum_{i>j}\left\|\nabla f_t^i - \nabla f_t^j\right\|^2 + \left\|\frac{1}{p}\sum_{i=1}^{p}\nabla f_t^i\right\|^2.
\end{aligned} \tag{58}$$

By the Cauchy-Schwarz inequality, we have

$$\frac{\mu L}{\mu + L} \left\| x_t^i - x_t^j \right\|^2 + \frac{1}{\mu + L} \left\| \nabla f_t^i - \nabla f_t^j \right\|^2 \geq \frac{2\sqrt{\mu L}}{\mu + L} \left\langle \nabla f_t^i - \nabla f_t^j, x_t^i - x_t^j \right\rangle. \tag{59}$$

Combining the above estimates in Equations 57, 58, 59, we obtain

$$
\begin{aligned}
\|y_{t+1} - x^*\|^2 &\leq \|y_t - x^*\|^2 + \eta^2 \left\| \frac{1}{p} \sum_{i=1}^{p} \nabla f_t^i - \xi_t \right\|^2 + \alpha^2 \|y_t - \tilde{x}_t\|^2 \\
&\quad - 2\eta \left( \frac{\mu L}{\mu + L} \|y_t - x^*\|^2 + \frac{1}{\mu + L} \left\| \frac{1}{p} \sum_{i=1}^{p} \nabla f_t^i \right\|^2 \right) \\
&\quad + 2\eta \left( 1 - \frac{2\sqrt{\mu L}}{\mu + L} \right) \frac{1}{p^2} \sum_{i>j} \left\langle \nabla f_t^i - \nabla f_t^j, x_t^i - x_t^j \right\rangle \\
&\quad + 2\eta \left\langle \xi_t, y_t - x^* \right\rangle - \alpha(\|y_t - x^*\|^2 + \|y_t - \tilde{x}_t\|^2 - \|\tilde{x}_t - x^*\|^2) \\
&\quad + 2\eta\alpha \left\langle \frac{1}{p} \sum_{i=1}^{p} \nabla f_t^i - \xi_t, y_t - \tilde{x}_t \right\rangle. \tag{60}
\end{aligned}
$$

Similarly if $0 \leq \alpha < 1$, we can have this upper-bound for the terms $\alpha^2 \|y_t - \tilde{x}_t\|^2 - \alpha \|y_t - \tilde{x}_t\|^2 + 2\eta\alpha \left\| \frac{1}{p} \sum_{i=1}^{p} \nabla f_t^i \right\| \|y_t - \tilde{x}_t\| \leq \frac{\eta^2 \alpha}{1-\alpha} \left\| \frac{1}{p} \sum_{i=1}^{p} \nabla f_t^i \right\|^2$ by applying $-ax^2 + bx \leq \frac{b^2}{4a}$ with $x = \|y_t - \tilde{x}_t\|$. Thus we have the following bound for the Equation 60

$$
\begin{aligned}
\|y_{t+1} - x^*\|^2 &\leq \left( 1 - 2\eta \frac{\mu L}{\mu + L} - \alpha \right) \|y_t - x^*\|^2 + \left( \eta^2 + \frac{\eta^2 \alpha}{1-\alpha} - \frac{2\eta}{\mu + L} \right) \left\| \frac{1}{p} \sum_{i=1}^{p} \nabla f_t^i \right\|^2 \\
&\quad - 2\eta^2 \left\langle \frac{1}{p} \sum_{i=1}^{p} \nabla f_t^i, \xi_t \right\rangle + 2\eta \left\langle \xi_t, y_t - x^* \right\rangle - 2\eta\alpha \left\langle \xi_t, y_t - \tilde{x}_t \right\rangle \\
&\quad + 2\eta \left( 1 - \frac{2\sqrt{\mu L}}{\mu + L} \right) \frac{1}{p^2} \sum_{i>j} \left\langle \nabla f_t^i - \nabla f_t^j, x_t^i - x_t^j \right\rangle \\
&\quad + \eta^2 \|\xi_t\|^2 + \alpha \|\tilde{x}_t - x^*\|^2. \tag{61}
\end{aligned}
$$

Since $\frac{2\sqrt{\mu L}}{\mu + L} \leq 1$, we need also bound the non-linear term $\left\langle \nabla f_t^i - \nabla f_t^j, x_t^i - x_t^j \right\rangle \leq L \left\| x_t^i - x_t^j \right\|^2$. Recall the bias-variance relation $\frac{1}{p} \sum_{i=1}^{p} \left\| x_t^i - x^* \right\|^2 = \frac{1}{p^2} \sum_{i>j} \left\| x_t^i - x_t^j \right\|^2 + \|y_t - x^*\|^2$. The key observation is that if $\frac{1}{p} \sum_{i=1}^{p} \left\| x_t^i - x^* \right\|^2$ remains bounded, then larger variance $\sum_{i>j} \left\| x_t^i - x_t^j \right\|^2$ implies smaller bias $\|y_t - x^*\|^2$. Thus this non-linear term can be compensated.

Again choose $\eta$ small enough such that $\eta^2 + \frac{\eta^2 \alpha}{1-\alpha} - \frac{2\eta}{\mu + L} \leq 0$ and take expectation in Equation 61,

$$
\begin{aligned}
\mathbb{E} \|y_{t+1} - x^*\|^2 &\leq \left( 1 - 2\eta \frac{\mu L}{\mu + L} - \alpha \right) \mathbb{E} \|y_t - x^*\|^2 \\
&\quad + 2\eta L \left( 1 - \frac{2\sqrt{\mu L}}{\mu + L} \right) \left( \frac{1}{p} \sum_{i=1}^{p} \mathbb{E} \left\| x_t^i - x^* \right\|^2 - \mathbb{E} \|y_t - x^*\|^2 \right) \\
&\quad + \eta^2 \frac{\sigma^2}{p} + \alpha \mathbb{E} \|\tilde{x}_t - x^*\|^2. \tag{62}
\end{aligned}
$$

As for the center variable in Equation 41, we apply simply the convexity of the norm $\|\cdot\|^2$ to obtain

$$\|\tilde{x}_{t+1} - x^*\|^2 \leq (1 - \beta) \|\tilde{x}_t - x^*\|^2 + \beta \|y_t - x^*\|^2. \tag{63}$$

Combing the estimates from Equations 50, 62, 63, and denote $a_t = \mathbb{E}\left\|y_t - x^*\right\|^2$, $b_t = \frac{1}{p}\sum_{i=1}^{p}\mathbb{E}\left\|x_t^i - x^*\right\|^2$, $c_t = \mathbb{E}\left\|\tilde{x}_t - x^*\right\|^2$, $\gamma_1 = 2\eta\frac{\mu L}{\mu+L}$, $\gamma_2 = 2\eta L(1 - \frac{2\sqrt{\mu L}}{\mu+L})$, then

$$\begin{pmatrix} a_{t+1} \\ b_{t+1} \\ c_{t+1} \end{pmatrix} \leq \begin{pmatrix} 1 - \gamma_1 - \gamma_2 - \alpha & \gamma_2 & \alpha \\ 0 & 1 - \gamma_1 - \alpha & \alpha \\ \beta & 0 & 1 - \beta \end{pmatrix} \begin{pmatrix} a_t \\ b_t \\ c_t \end{pmatrix} + \begin{pmatrix} \eta^2\frac{\sigma^2}{p} \\ \eta^2\sigma^2 \\ 0 \end{pmatrix},$$

as long as $0 \leq \beta \leq 1$, $0 \leq \alpha < 1$ and $\eta^2 + \frac{\eta^2\alpha}{1-\alpha} - \frac{2\eta}{\mu+L} \leq 0$, i.e. $0 \leq \eta \leq \frac{2}{\mu+L}(1-\alpha)$.  □

# 8 Additional pseudo-codes of the algorithms

## 8.1 DOWNPOUR pseudo-code

Algorithm 3 captures the pseudo-code of the implementation of the DOWNPOUR used in this paper.

---

**Algorithm 3:** DOWNPOUR: Processing by worker $i$ and the master

**Input:** learning rate $\eta$, communication period $\tau \in \mathbb{N}$
**Initialize:** $\tilde{x}$ is initialized randomly, $x^i = \tilde{x}$, $v^i = 0$, $t^i = 0$

**Repeat**
  **if** ($\tau$ divides $t^i$) **then**
    $\tilde{x} \leftarrow \tilde{x} + v^i$
    $x^i \leftarrow \tilde{x}$
    $v^i \leftarrow 0$
  **end**
  $x^i \leftarrow x^i - \eta g_{t^i}^i(x^i)$
  $v^i \leftarrow v^i - \eta g_{t^i}^i(x^i)$
  $t^i \leftarrow t^i + 1$
**Until forever**

---

## 8.2 MDOWNPOUR pseudo-code

Algorithms 4 and 5 capture the pseudo-codes of the implementation of momentum DOWNPOUR (MDOWNPOUR) used in this paper. Algorithm 4 shows the behavior of each local worker and Algorithm 5 shows the behavior of the master.

---

**Algorithm 4:** MDOWNPOUR: Processing by worker $i$

**Initialize:** $x^i = \tilde{x}$

**Repeat**
  Receive $\tilde{x}$ from the master: $x^i \leftarrow \tilde{x}$
  Compute gradient $g^i = g^i(x^i)$
  Send $g^i$ to the master
**Until forever**

---

**Algorithm 5:** MDOWNPOUR: Processing by the master

**Input:** learning rate $\eta$, momentum term $\delta$
**Initialize:** $\tilde{x}$ is initialized randomly, $v^i = 0$,

**Repeat**
  Receive $g^i$
  $v \leftarrow \delta v - \eta g^i$
  $\tilde{x} \leftarrow \tilde{x} + \delta v$
**Until forever**

---

# 9 Experiments - additional material

## 9.1 Data preprocessing

For the *ImageNet* experiment, we re-size each RGB image so that the smallest dimension is $256$ pixels. We also re-scale each pixel value to the interval $[0, 1]$. We then extract random crops (and their horizontal flips) of size $3 \times 221 \times 221$ pixels and present these to the network in mini-batches of size $128$.

For the *CIFAR* experiment, we use the original RGB image of size $3 \times 32 \times 32$. As before, we re-scale each pixel value to the interval $[0, 1]$. We then extract random crops (and their horizontal flips) of size $3 \times 28 \times 28$ pixels and present these to the network in mini-batches of size $128$.

The training and test loss and the test error are only computed from the center patch ($3 \times 28 \times 28$) for the *CIFAR* experiment and the center patch ($3 \times 221 \times 221$) for the *ImageNet* experiment.

## 9.2 Data prefetching (Sampling the dataset by the local workers)

We will now explain precisely how the dataset is sampled by each local worker as uniformly and efficiently as possible. The general parallel data loading scheme on a single machine is as follows: we use $k$ CPUs, where $k = 8$, to load the data in parallel. Each data loader reads from the memory-mapped (mmap) file a chunk of $c$ raw images (preprocessing was described in the previous subsection) and their labels (for *CIFAR* $c = 512$ and for *ImageNet* $c = 64$). For the *CIFAR*, the mmap file of each data loader contains the entire dataset whereas for *ImageNet*, each mmap file of each data loader contains different $1/k$ fractions of the entire dataset. A chunk of data is always sent by one of the data loaders to the first worker who requests the data. The next worker requesting the data from the same data loader will get the next chunk. Each worker requests in total $k$ data chunks from $k$ different data loaders and then process them before asking for new data chunks. Notice that each data loader cycles through the data in the mmap file, sending consecutive chunks to the workers in order in which it receives requests from them. When the data loader reaches the end of the mmap file, it selects the address in memory uniformly at random from the interval $[0, s]$, where $s = (\text{number of images in the mmap file modulo mini-batch size})$, and uses this address to start cycling again through the data in the mmap file. After the local worker receives the $k$ data chunks from the data loaders, it shuffles them and divides it into mini-batches of size $128$.

## 9.3 Learning rates

In Table 1 we summarize the learning rates $\eta$ (we used constant learning rates) explored for each method shown in Figure 2. For all values of $\tau$ the same set of learning rates was explored for each method.

Table 1: Learning rates explored for each method shown in Figure 2 (*CIFAR* experiment).

|  | $\eta$ |
|---|---|
| EASGD | $\{0.05, 0.01, 0.005\}$ |
| EAMSGD | $\{0.01, 0.005, 0.001\}$ |
| DOWNPOUR ADOWNPOUR MVADOWNPOUR | $\{0.005, 0.001, 0.0005\}$ |
| MDOWNPOUR | $\{0.00005, 0.00001, 0.000005\}$ |
| SGD, ASGD, MVASGD | $\{0.05, 0.01, 0.005\}$ |
| MSGD | $\{0.001, 0.0005, 0.0001\}$ |

In Table 2 we summarize the learning rates $\eta$ (we used constant learning rates) explored for each method shown in Figure 3. For all values of $p$ the same set of learning rates was explored for each method.

In Table 3 we summarize the initial learning rates $\eta$ we use for each method shown in Figure 4. For all values of $p$ the same set of learning rates was explored for each method. We also used the rule of the thumb to decrease the initial learning rate twice, first time we divided it by $5$ and the second time by $2$, when we observed that the decrease of the online predictive (training) loss saturates.

Table 2: Learning rates explored for each method shown in Figure 3 (*CIFAR* experiment).

|  | $\eta$ |
|---|---|
| EASGD | $\{0.05, 0.01, 0.005\}$ |
| EAMSGD | $\{0.01, 0.005, 0.001\}$ |
| DOWNPOUR | $\{0.005, 0.001, 0.0005\}$ |
| MDOWNPOUR | $\{0.00005, 0.00001, 0.000005\}$ |
| SGD, ASGD, MVASGD | $\{0.05, 0.01, 0.005\}$ |
| MSGD | $\{0.001, 0.0005, 0.0001\}$ |

Table 3: Learning rates explored for each method shown in Figure 4 (*ImageNet* experiment).

|  | $\eta$ |
|---|---|
| EASGD | 0.1 |
| EAMSGD | 0.001 |
| DOWNPOUR | for $p = 4$: 0.02<br>for $p = 8$: 0.01 |
| SGD, ASGD, MVASGD | 0.05 |
| MSGD | 0.0005 |

## 9.4 Comparison of *SGD*, *ASGD*, *MVASGD* and *MSGD*

Figure 6: Convergence of the training and test loss (negative log-likelihood) and the test error (original and zoomed) computed for the center variable as a function of wallclock time for *SGD*, *ASGD*, *MVASGD* and *MSGD* ($p = 1$) on the *CIFAR* experiment.

Figure 7: Convergence of the training and test loss (negative log-likelihood) and the test error (original and zoomed) computed for the center variable as a function of wallclock time for *SGD*, *ASGD*, *MVASGD* and *MSGD* ($p = 1$) on the *ImageNet* experiment.

Figure 6 shows the convergence of the training and test loss (negative log-likelihood) and the test error computed for the center variable as a function of wallclock time for *SGD*, *ASGD*, *MVASGD* and *MSGD* ($p = 1$) on the *CIFAR* experiment. For all *CIFAR* experiments we always start the averaging for the $ADOWNPOUR$ and $ASGD$ methods from the very beginning of each experiment. For all *ImageNet* experiments we start the averaging for the $ASGD$ at the same time when we first reduce the initial learning rate.

Figure 7 shows the convergence of the training and test loss (negative log-likelihood) and the test error computed for the center variable as a function of wallclock time for *SGD*, *ASGD*, *MVASGD* and *MSGD* ($p = 1$) on the *ImageNet* experiment.

## 9.5 Dependence of the learning rate

This section discusses the dependence of the trade-off between exploration and exploitation on the learning rate. We compare the performance of respectively *EAMSGD* and *EASGD* for different learning rates $\eta$ when $p = 16$ and $\tau = 10$ on the *CIFAR* experiment. We observe in Figure 8 that higher learning rates $\eta$ lead to better test performance for the *EAMSGD* algorithm which potentially can be justified by the fact that they sustain higher fluctuations of the local workers. We conjecture that higher fluctuations lead to more exploration and simultaneously they also impose higher regularization. This picture however seems to be opposite for the *EASGD* algorithm for which larger learning rates hurt the performance of the method and lead to overfitting. Interestingly in this experiment for both *EASGD* and *EAMSGD* algorithm, the learning rate for which the best training performance was achieved simultaneously led to the worst test performance.

Figure 8: Convergence of the training loss (negative log-likelihood, original) and the test error (zoomed) computed for the center variable as a function of wallclock time for *EAMSGD* and *EASGD* run with different values of $\eta$ on the *CIFAR* experiment. $p = 16, \tau = 10$.

## 9.6 Dependence of the communication period

This section discusses the dependence of the trade-off between exploration and exploitation on the communication period. We have observed from the *CIFAR* experiment that *EASGD* algorithm exhibits very similar convergence behavior when $\tau = 1$ up to even $\tau = 1000$, whereas *EAMSGD* can get trapped at worse energy (loss) level for $\tau = 100$. This behavior of *EAMSGD* is most likely due to the non-convexity of the objective function. Luckily, it can be avoided by gradually decreasing the learning rate, i.e. increasing the penalty term $\rho$ (recall $\alpha = \eta\rho$), as shown in Figure 9. In contrast, the *EASGD* algorithm does not seem to get trapped at all along its trajectory. The performance of *EASGD* is less sensitive to increasing the communication period compared to *EAMSGD*, whereas for the *EAMSGD* the careful choice of the learning rate for large communication periods seems crucial.

Compared to all earlier results, the experiment in this section is re-run three times with a new random[12] seed and with faster cuDNN[13] package[14]. All our methods are implemented in Torch[15]. The Message Passing Interface implementation MVAPICH2[16] is used for the GPU-CPU communication.

Figure 9: Convergence of the training loss (negative log-likelihood, original) and the test error (zoomed) computed for the center variable as a function of wallclock time for *EASGD* and *EAMSGD* ($p = 16, \eta = 0.01, \beta = 0.9, \delta = 0.99$) on the *CIFAR* experiment with various communication period $\tau$ and learning rate decay $\gamma$. The learning rate is decreased gradually over time based each local worker's own clock $t$ with $\eta_t = \eta/(1 + \gamma t)^{0.5}$.

## 9.7 Breakdown of the wallclock time

In addition, we report in Table 4 the breakdown of the total running time for *EASGD* when $\tau = 10$ (the time breakdown for *EAMSGD* is almost identical) and *DOWNPOUR* when $\tau = 1$ into computation time, data loading time and parameter communication time. For the *CIFAR* experiment the reported time corresponds to processing $400 \times 128$ data samples whereas for the *ImageNet* experiment it corresponds to processing $1024 \times 128$ data samples. For $\tau = 1$ and $p \in \{8, 16\}$ we observe that the communication time accounts for significant portion of the total running time whereas for $\tau = 10$ the communication time becomes negligible compared to the total running time (recall that based on previous results *EASGD* and *EAMSGD* achieve best performance with larger $\tau$ which is ideal in the setting when communication is time-consuming).

| | $p=1$ | $p=4$ | $p=8$ | $p=16$ |
|---|---|---|---|---|
| $\tau = 1$ | 12/1/0 | 11/2/3 | 11/2/5 | 11/2/9 |
| $\tau = 10$ | NA | 11/2/1 | 11/2/1 | 12/2/1 |

| | $p=1$ | $p=4$ | $p=8$ |
|---|---|---|---|
| $\tau = 1$ | 1248/20/0 | 1323/24/173 | 1239/61/284 |
| $\tau = 10$ | NA | 1254/58/7 | 1266/84/11 |

Table 4: Approximate computation time, data loading time and parameter communication time [sec] for *DOWNPOUR* (top line for $\tau = 1$) and *EASGD* (the time breakdown for *EAMSGD* is almost identical) (bottom line for $\tau = 10$). Left time corresponds to *CIFAR* experiment and right table corresponds to *ImageNet* experiment.

## 9.8 Time speed-up

In Figure 10 and 11, we summarize the wall clock time needed to achieve the same level of the test error for all the methods in the *CIFAR* and *ImageNet* experiment as a function of the number of local workers $p$. For the *CIFAR* (Figure 10) we examined the following levels: $\{21\%, 20\%, 19\%, 18\%\}$ and for the *ImageNet* (Figure 11) we examined: $\{49\%, 47\%, 45\%, 43\%\}$. If some method does not appear on the figure for a given test error level, it indicates that this method never achieved this level. For the *CIFAR* experiment we observe that from among *EASGD*, *DOWNPOUR* and *MDOWNPOUR* methods, the *EASGD* method needs less time to achieve a particular level of test error. We observe that with higher $p$ each of these methods does not necessarily need less time to achieve the same level of test error. This seems counter intuitive though recall that the learning rate for the methods is selected based on the smallest achievable test error. For larger $p$ smaller learning rates were selected than for smaller $p$ which explains our results. Meanwhile, the *EAMSGD* method achieves significant speed-up over other methods for all the test error levels. For the *ImageNet* experiment we observe that all methods outperform *MSGD* and furthermore with $p = 4$ or $p = 8$ each of these methods requires less time to achieve the same level of test error. The *EAMSGD* consistently needs less time than any other method, in particular *DOWNPOUR*, to achieve any of the test error levels.

Figure 10: The wall clock time needed to achieve the same level of the test error $thr$ as a function of the number of local workers $p$ on the *CIFAR* dataset. From left to right: $thr = \{21\%, 20\%, 19\%, 18\%\}$. Missing bars denote that the method never achieved specified level of test error.

Figure 11: The wall clock time needed to achieve the same level of the test error $thr$ as a function of the number of local workers $p$ on the *ImageNet* dataset. From left to right: $thr = \{49\%, 47\%, 45\%, 43\%\}$. Missing bars denote that the method never achieved specified level of test error.

.

## Footnotes

[10]In our notation, $\mathbb{V}$ denotes the variance.

[11]As a side note, notice that the center parameter $\tilde{x}_t$ is tracking the spatial average $y_t$ of the local parameters with a non-symmetric spring in Equation 31 and 32. To be more precise note that the update on $y_{t+1}$ contains $(\tilde{x}_t - y_t)$ scaled by $\alpha$, whereas the update on $\tilde{x}_{t+1}$ contains $-(\tilde{x}_t - y_t)$ scaled by $\beta$. Since $\alpha = \beta/p$ the impact of the center $\tilde{x}_{t+1}$ on the spatial local average $y_{t+1}$ becomes more negligible as $p$ grows.

[12]To clarify, the random initialization we use is by default in Torch's implementation.

[13]https://developer.nvidia.com/cuDNN

[14]https://github.com/soumith/cudnn.torch

[15]http://torch.ch

[16]http://mvapich.cse.ohio-state.edu