[Reviews · NeurIPS 2015]

Submitted by Assigned_Reviewer_1

In this paper, the authors propose a distributed stochastic gradient learning scheme amenable to infrequent synchronization. Their claim is that, due to the non-convexity, existing algorithms are not very robust to distant synchronizations. Further, their resilience to differences between parameter values across different nodes favors exploration, which leads to better solutions.

The algorithm is simple and seems to work well on the experiments. The authors compared with many existing algorithms and for several values of tau. To nitpick, since the authors talk about deep learning without more details, I would also have appreciated experiments on other architectures than CNNs.

My main gripe with this paper is that, while they revisit an old method (see section 4.1 of "Notes on Big-n Problems" by Mark Schmidt, an excellent review of the literature), their coverage of existing implementations of these methods is scarce. Basically, they are referenced in the introduction but the in-depth analysis is limited to ADMM.

I believe positive results on distributed stochastic optimization in the context of nonconvex losses can definitely be of great significance. However, the authors should do a better job at reviewing the existing literature and not just referencing it.
Summary: The algorithm seems to outperform existing distributed techniques but is very slim on the comparisons with non-deep learning oriented distribution techniques which have been widely studies in the optimization literature.

Submitted by Assigned_Reviewer_2

This paper studies general-purpose training algorithms for deep learning and proposes a family of algorithms called elastic averaging SGD. The idea is novel and the paper is of very high quality.

The paper focuses on training large-scale deep learning models under communication constraints. This problem is difficult since there are many local optima in non-convex problems like in deep learning. The optimization problem is formulated as a global variable consensus problem such that local workers would not fall into different local optima, and then its gradient update rules are reinterpreted using the elastic forces between local and global parameters. This is why the proposed family of algorithms is named elastic averaging SGD (EASGD). There are four algorithms in the family: a synchronous version, an asynchronous version, and their momentum versions. They are demonstrated promising theoretically and experimentally.

In the theoretical analysis, the authors present a vivid illustrative example in which the popular ADMM can be unstable and its stability condition is unknown, but the stability condition of EASGD is quite simple. The writing style here is very impressive and friendly. All theorems are in the supplementary material and there is only a small part of theoretical analysis in the main paper, but the organization made me feel quite smooth (though I did not carefully go through the supplementary material). I have some minor questions here: What is the most important reason for the superior stability of EASGD? Is it because the maps of ADMM are asymmetric while the maps of EASGD are symmetric? Is it possible to find another illustrative example such that on this specific example EASGD is more unstable than ADMM?

In their experiments, the proposed algorithms were used to train convolutional neural networks and they outperformed state-of-the-art SGD algorithms on CIFAR and ImageNet. The proposed algorithms achieved the goal such that the larger the communication periods are, the better EASGD would be than other SGD as shown in Figure 2. Figures 3 and 4 also showed that the speedup of EASGD would be better than the baselines when there were more workers. As further experimental results, we have implemented similar gradient update rules adapted to our clusters (according to the arxiv version) and they successfully improved our baselines. The resulted models have been or will be launched online and one fifth of the world's population would benefit from these improved models. So I believe that this is a significant paper.
Summary: This paper studies general-purpose training algorithms for deep learning and proposes a family of algorithms called elastic averaging SGD. The idea is novel and the paper is of very high quality.

Submitted by Assigned_Reviewer_3

The paper addresses the important problem of parallel optimization. The proposed algorithm is shown to have better tolerance to staleness and thus carries potentially smaller communication burden than the state-of-the-art. It also obtains better learning results as measured by test error. The paper is overall well-written and a pleasure to read.

In the current EASGD, the center variable is updated by following a symmetric elastic force (Eq.4). However given the objective in Eq.2, a more natural solution seems to be always taking the exact average of all the local variables (still in an online fashion). I wonder whether the authors have investigated this variant and would like to see some discussion in the paper.

Is there any experimental study on the effect of rho? More specifically, how does the exploration affect the performance?

the line under Eq.4: the stochastic gradient of "F"?
Summary: The paper proposes a parallel algorithm that encourages simultaneous exploration among computing nodes for more effectively optimizing objectives with many local optima. The proposed idea is well motivated, clearly presented and supported by extensive experimental studies.

Submitted by Assigned_Reviewer_4

The paper introduces EAMSGD algorithm for fast convergence of distributed SGD. The idea is based off several prior class works in distributed optimization and is first drawn a connection with deep learning models. Results on public datasets CIFAR and ImageNet are solid and shown clearly.

Quality: high quality paper with good ideas and solid results.

Clarity: the paper is well structured, at the same time, clearer and more direct formulations of the algorithm might be more appreciated by the reader. For instance, the different types of distributed sgd at start of section 5 is not clearly documented. Suggest a table format or show the list of available algorithms with clear categories. It will bring out the results even better.

Originality: since the idea is draw from prior classic works, the idea is not entirely new, at the same time the application to recent deep learning models and dataset is a first-time.

Significance: due to the flourishing of distributed deep learning in industry and academia, this paper should be appreciated by many engineers and researchers.

Summary: The paper, with the proposed elastic averaging method for distributed SGD, shows good results on empirical image classification experiments. The authors also provided theoretical analysis and mathematical formulations by drawing analogy to ADMM. The research idea is novel and results are solid and show clearly. Other than some details in experiments which could be improved, this is a good paper recommended to be published at NIPS.

Author Feedback
Author rebuttal: We thank the Reviewers for their valuable feedback. We are thrilled to read that Rev_1 re-implemented our approach and finds it useful and beneficial to "one fifth of the world's population". We introduced all specific comments into our current draft and respond below.

Rev_1
We are thrilled to receive such a positive feedback. Reviewer also shows correct and thoughtful understanding of our approach. We are pleased to see that the Reviewer re-implemented our work and finds it extremely influential.
The reason why ADMM is unstable is due to the Lagrangian update, which contributes to the linear map's asymmetry. But it also makes the dynamical system oscillate by introducing complex-valued eigenvalues when the penalty \rho is small. In case of EASGD, the map is symmetric, thus the eigenvalues are always real-valued and therefore no such oscillation is possible. The example where EASGD could potentially be unstable might exist in the asynchronous case, though it is highly non-trivial in our opinion to construct such example, and we never observed better stability of ADMM over EASGD in our experiments as we also state in the paper.
Rev_2
The update suggested by the Reviewer is a very special case of Eq. 4 when \beta =1 (and \alpha and \beta are potentially decoupled). In case when \beta is set to \beta = 1 one may obtain inferior performance compared to smaller values of \beta as is captured in Figure 5 and Corollary 7.1 in the Supplement.
Our experiments study the effect of \rho extensively and thus the connection between exploration and performance. Recall that in the asynchronous case effective \beta (see Footnote 7) is equal to \beta/\tau and thus \rho = \beta/(\tau \eta p). We explore different settings of \rho by changing \tau, \eta, and p.
Rev_4
The vast majority of GPU cycles in case of training deep learning systems are consumed with training CNNs. Those are also the most challenging system components to study. See also last paragraph of Rev_1's comments.
Our paper is targeting deep learning. We compare with techniques that were successfully applied to deep learning before (DOWNPOUR) and techniques that are closely-related to ours (Stochastic ADMM). Regarding Schmidt 2012 and Mahajan et al 2015: our setting is fundamentally different. Note that in our paper (i) we do not consider distributing the data among local workers (as the data communication is not the bottleneck), thus each local worker samples the same non-convex function, and (ii) we address small \rho regime (as the parameter size can be very large: it is 1.3MB for the CIFAR and 233MB for the ImageNet): we encourage local workers to explore more, i.e. fluctuate further from the center. We compare all methods in this setting. It is unclear to us how to adapt Mahajan et al 2015 to our setting (e.g. they need to compute the full gradient which is non-trivial due to the dropout regularization). Also Sect. 4.1. in Schmidt 2012 suggests that ADMM is the most closely related approach to our approach.
Rev_5
In the Supplement (Sect. 9.7) we show the breakdown of the wallclock time into computation time, data loading time and parameter communication time for different settings of p. Already for \tau = 10 the difference in parameter communication time between single node case (p = 1) and distributed case (p > 4) is clearly negligible, i.e. as p grows the parameter communication time stays nearly unchanged. The network is therefore not saturated for \tau = 10, thus running p=16 on 4 machines with a fixed synchronization frequency would have a similar performance as p=16 on 16 machines (use one gpu per machine). Also, for all our experiments we use a GPU-cluster interconnected with InfiniBand. There exists two large-scale learning systems: GPU-based and CPU-based. GPU-based system has smaller number of nodes, typically a cluster has 8 nodes (the supercomputer Minwa has 36 GPU nodes, see http://arxiv.org/pdf/1501.02876v1.pdf) with 4 GPUs each. CPU-based system relying on massive parallelization can be converted into GPU-based system. Also, universities do not have access to the infrastructure that would enable the experiments with tens of thousands of CPU cores. Regarding superiority of our method over other baselines: we demonstrate it in the paper but please see also the last paragraph of Rev_1's comments. Rev_1 re-implemented our algorithms and find its superior to other baselines and beneficial to real-life important applications.
Rev_6
Our stability analysis in the first half of the paper (Sect. 4) addresses the behavior of the algorithm near a stationary point (the local behavior), whereas the function can be globally non-convex. The second part of the paper is largely experimental and shows the global behavior of the algorithm in the non-convex setting. The algorithm is shown to be successful in this setting. Also note that the problem of providing theoretical guarantees in non-convex setting is largely open.